# Cathepsin D deficiency in mammary epithelium transiently stalls breast cancer by interference with mTORC1 signaling

Stephanie Ketterer[1,2,3], Julia Mitschke[1], Anett Ketscher[1], Manuel Schlimpert[2,4,5], Wilfried Reichardt [3,6], Natascha Baeuerle[1], Maria Elena Hess[1,2,7,8], Patrick Metzger [1,2,7], Melanie Boerries [1,3,7,8], Christoph Peters[1,3,8,9], Bernd Kammerer[5,9], Tilman Brummer [1,3,9], Florian Steinberg[5] & Thomas Reinheckel [1,3,8,9 ✉]

Cathepsin D (CTSD) is a lysosomal protease and a marker of poor prognosis in breast cancer. However, the cells responsible for this association and the function of CTSD in cancer are still incompletely understood. By using a conditional CTSD knockout mouse crossed to the transgenic *MMTV-PyMT* breast cancer model we demonstrate that CTSD deficiency in the mammary epithelium, but not in myeloid cells, blocked tumor development in a cell-autonomous manner. We show that lack of CTSD impaired mechanistic Target of Rapamycin Complex 1 (mTORC1) signaling and induced reversible cellular quiescence. In line, CTSD-deficient tumors started to grow with a two-month delay and quiescent *Ctsd^-/-* tumor cells restarted proliferation upon long-term culture. This was accompanied by rewiring of oncogenic gene expression and signaling pathways, while mTORC1 signaling remained permanently disabled in CTSD-deficient cells. Together, these studies reveal a tumor cell-autonomous effect of CTSD deficiency, and establish a pivotal role of this protease in the cellular response to oncogenic stimuli.

[1] Institute of Molecular Medicine and Cell Research, University of Freiburg, Freiburg 79104, Germany. [2] Faculty of Biology, University of Freiburg, Freiburg 79104, Germany. [3] German Cancer Consortium (DKTK) and German Cancer Research Center (DKFZ), partner site Freiburg, Freiburg 79104, Germany. [4] Spemann Graduate School of Biology and Medicine, University of Freiburg, Freiburg 79104, Germany. [5] Centre for Integrative Signalling Analysis (CISA), University of Freiburg, Freiburg 79104, Germany. [6] Medical Physics, Department of Radiology, University Medical Center Freiburg, Freiburg 79106, Germany. [7] Institute of Medical Bioinformatics and Systems Medicine, Medical Center – University of Freiburg, Faculty of Medicine, University of Freiburg, Freiburg 79110, Germany. [8] Comprehensive Cancer Center Freiburg (CCCF), Medical Center – University of Freiburg, Faculty of Medicine, University of Freiburg, Freiburg 79106, Germany. [9] BIOSS Centre for Biological Signalling Studies, University of Freiburg, Freiburg 79104, Germany. ✉email: thomas.reinheckel@mol-med.uni-freiburg.de

athepsin D (CTSD) is a major cellular endopeptidase with primary location in the endosomal/lysosomal compartment[1]. Unlike most other lysosomal proteases dubbed cathepsin, which mostly rely on cysteine residues for cleaving peptide bonds, CTSD is using an aspartate for catalysis, which classifies CTSD as aspartic proteinase. Although mainly located and active in the acidic cell compartment, CTSD has been found to be secreted into the extracellular milieu and, in addition, to be transferred from the lysosome to the cytosol or the nucleus of cells where it has been implicated in cell death processes and gene regulation, respectively[2,3]. Interestingly, and again in contrast to cysteine cathepsins, no bona fide CTSD protease inhibitors exist extracellularly or in the cytosol. Therefore, the role of CTSD in remodeling the extracellular matrix was studied under physiological conditions and in various disease states[4,5]. Also the inactive zymogen of CTSD (the so-called pro-CTSD) is secreted and was shown to be mitogenic through interaction with several cell surface receptors[6].

Based on its generally high and ubiquitous expression, as well as its variable cellular localization, CTSD has raised much interest in studies concerning its role in diseases, with solid cancers being a prominent example for CTSD gain of function[6]. In contrast to its excess, the lack of CTSD is also detrimental. Genetic deficiency of CTSD causes neuronal ceroid-lipofuscinosis type 10 (CLN10), a progressive hereditary neurodegenerative disease[7,8]. Patients with congenital CLN10, caused by complete absence of CTSD activity, die within days after birth. In line with the human disease, mice deficient for CTSD develop a lysosomal storage disorder and neuronal death before they die prematurely around 26 days after birth[9,10].

The early death of mice with constitutive deficiency of CTSD has considerably complicated the analysis of CTSD functions in animal models of human disease. As a result, research on CTSD in tumor biology is lagging far behind that on cysteine-type cathepsins. Those have been readily examined in animal models for their potential use as drug targets, diagnostic imaging agents, and prodrug activators, even though there is functional redundancy and compensatory mechanisms between cysteine cathepsin family members[11,12]. This backlog is hapless, because CTSD has a long-standing tradition of being associated with progression of cancer, especially breast cancer. Already in 1980, a glycoprotein of 52 kDa size was described that is synthesized and secreted by the estrogen receptor (ER)-positive human breast cancer cell line MCF-7 in response to estradiol treatment[13]. This protein was later identified as pro-CTSD and found to be secreted constitutively from triple-negative breast cancer (TNBC) cell lines (BT-20, MDA-MB231)[14,15]. This cancer cell-specific secretion of pro-CTSD can be targeted by antibodies to elicit an anti-tumor immune response[16]. Pro-CTSD applied to human breast cancer cell cultures promotes proliferation in vitro and its down-regulation decreased xenograft growth and experimental lung metastasis in mice[6]. In human breast cancer, CTSD was suggested as a tumor marker long time ago[17]. Since then, many investigations aimed to link CTSD protein levels or activities to the clinical outcome of breast cancer patients. The most comprehensive study investigated tumor samples from 2810 breast cancer patients and uncovered a positive correlation of high CTSD protein levels in tumor homogenates with cancer relapse and patient death[18]. Furthermore, the amount of CTSD was suggested to predict prognosis independent of any other clinical prognostic factor. Other studies clarified that CTSD levels in the tumor specimen, but not its proteolytic activity in patient serum, has prognostic value[19]. However, macrophages and stromal cells also express CTSD, and some research claims that there is no prognostic significance to cathepsin D expression in tumor cells, but in the stromal compartment of the tumor[20].

Genetic models allowing for cell-specific inactivation of CTSD are probably the best approach solving the aforementioned controversies on cell type-specific CTSD functions in breast cancer. Therefore, we set out to develop a conditional CTSD mouse model by means of the Cre-loxP technology. First, we reported that selective deletion of CTSD in the nervous system faithfully recapitulates the lethal CLN10 phenotype[21]. In the present study, we take advantage of this system by deleting CTSD selectively in mammary epithelial cells or in myeloid cells of the widely used transgenic *MMTV-PyMT* (PyMT) mouse model of human metastasizing breast cancer[22]. By this approach, we are able to avoid the CLN10 neurodegeneration and to show a marked delay of tumorigenesis upon CTSD deletion in mammary epithelial cells, while CTSD deficiency in myeloid cells does not affect tumor progression. To address the underlying mechanism, we generate tumor cell lines from mice with or without mammary epithelium-specific CTSD deficiency. Challenging CTSD-deficient PyMT cells by mild starvation induces quiescence, expands the acidic cell compartment, and increases autophagic flux. In addition, CTSD deficiency impairs mechanistic Target of Rapamycin Complex 1 (mTORC1) signaling, even under mTORC1-stimulating culture conditions. To our surprise, long-term starved CTSD-deficient tumor cells escape quiescence and start proliferating, as do tumors after a latency period of about two months. However, mTORC1 signaling is still perturbed in long-term starved CTSD-deficient cells. Instead, these cells upregulate compensatory oncogenic signaling pathways.

## Results

**Cell type-specific deletion of cathepsin D in murine mammary carcinoma.** Within breast cancer tissues epithelial cells and macrophages stain strongly for CTSD[23]. We set out to study cell type-specific functions of CTSD in breast cancer aided by transgenic PyMT mice. Mice harboring floxed *Ctsd* alleles[21] were crossed to *LysM-* or *MMTV-cre* deleter strains to specifically inactivate CTSD in myeloid and in mammary epithelial cells, respectively. Accordingly, bone marrow-derived macrophages of *LysM-cre;Ctsd*$^{-/-}$ mice lacked CTSD completely and tumors of *MMTV-cre;Ctsd*$^{-/-}$ PyMT mice showed strongly reduced CTSD levels (Supplementary Fig. 1a, b). The remaining CTSD in these tumors probably stems from non-epithelial stromal cells. A fluorescent cre reporter, switching from membrane-tagged Tomato (mT)- to Green Fluorescent Protein (mG)-expression in cells encountering a cre recombinase, was further used to visualize recombination in the mammary gland[24]. *MMTV-cre; mTmG* mice showed GFP expression in the basal and luminal layers of the mammary epithelium, but not in the surrounding adipose tissue, thereby demonstrating breast epithelium-specific recombination by the MMTV-cre recombinase (Supplementary Fig. 1c). Next, we tested whether CTSD deficiency in the mammary epithelium impairs normal breast function by analyzing the weaned-to-born ratio. There was no significant difference between offspring from *MMTV-cre;Ctsd*$^{-/-}$ and control mice indicating normal lactation (Supplementary Fig. 1d). In *MMTV-cre;Ctsd*$^{-/-}$ PyMT mice, expression of the cre recombinase and the PyMT oncogene are both under the control of the *MMTV* promoter. Quantitative RT-PCR for PyMT in tumors from such mice revealed that this did not lead to reduced mRNA expression of the oncogene when compared to control tumors (Supplementary Fig. 1e). As the *MMTV* promoter that controls PyMT expression is steroid hormone-driven, we examined the expression pattern of estrogen receptor α in normal and cancerous breast tissue. Epithelial cells showed a typical nuclear staining that did not differ in intensity or distribution between *MMTV-cre;Ctsd*$^{-/-}$ and control mice (Supplementary Fig. 1f).

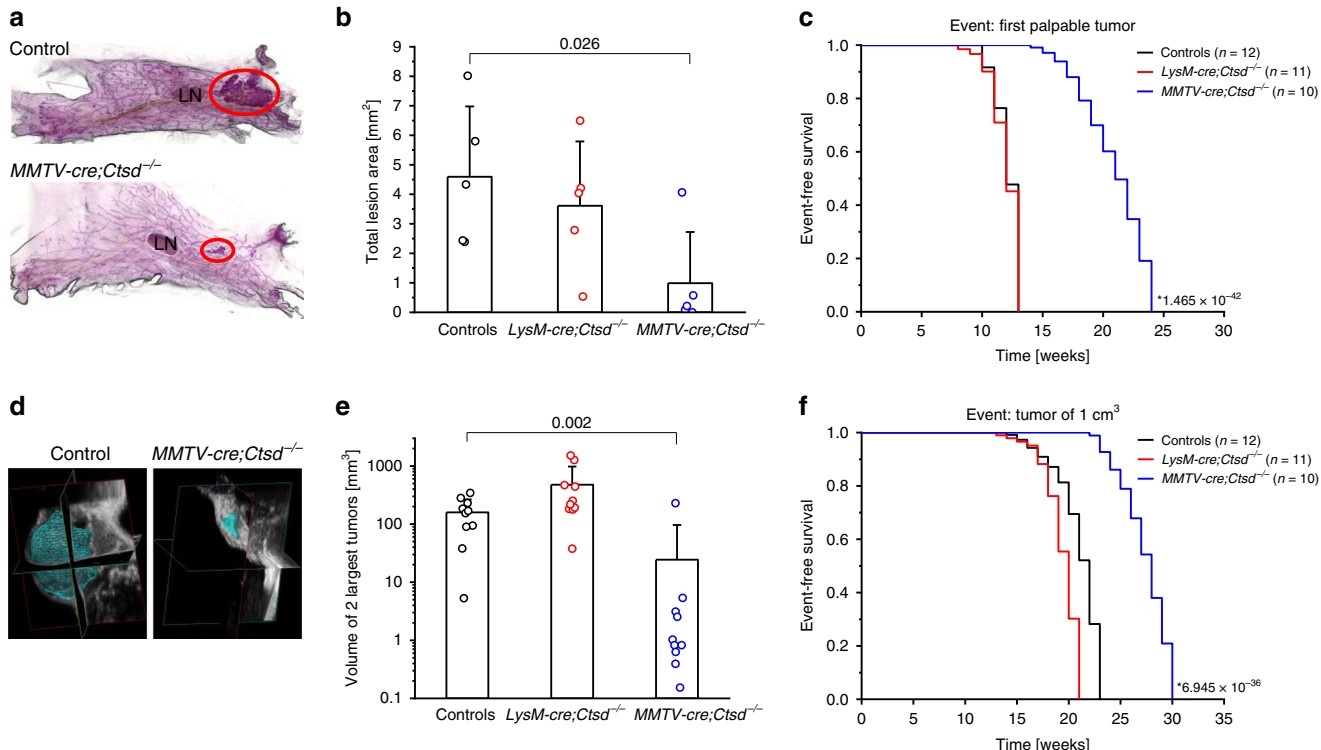

**Fig. 1 Mammary epithelium-specific CTSD deficiency delays tumor development in PyMT mice. a** Representative pictures of whole mounts of mammary glands from 8-week-old PyMT mice of indicated genotypes (Control: $Ctsd^{fl/fl}$). Areas occupied by dysplastic lesions are encircled in red and quantified in **b** ($n = 5$ animals; two-sided two-sample t-test). LN, lymph node. **c** Kaplan–Meier analysis for first palpable tumors in PyMT mice of indicated genotypes (Controls: $4\times Ctsd^{+/fl} + 8\times Ctsd^{fl/fl}$; *, two-sided log-rank test compared with controls). **d** Representative pictures and **e** quantification of volumetric ultrasound imaging of the two largest tumors detected per individual in 18-week-old PyMT mice of indicated genotypes (Controls: $2\times Ctsd^{+/fl} + 8\times Ctsd^{fl/fl}$; $n = 10$ animals; two-sided Mann–Whitney test). **f** Kaplan–Meier analysis for tumors at end-stage (1 cm$^3$) in PyMT mice of indicated genotypes (Controls: $4\times Ctsd^{+/fl} + 8\times Ctsd^{fl/fl}$; *, two-sided log-rank test compared with controls). Bar charts show all data points with mean + SD and p-value. Source data are provided as a Source Data file.

$C57/BL6\text{-}Tg(MMTV\text{-}PyMT)$ mice are more resistant to oncogene-induced transformation of the mammary epithelium compared to $FVB/N\text{-}Tg(MMTV\text{-}PyVT)634Mul/J$, resulting in palpable tumors only at around 13 weeks of age in the C57/BL6 background[25]. To assess early tumorigenesis, we quantified dysplastic lesions in whole mounts of mammary glands obtained from 8-week-old tumor mice. The lesion burden was similar for control and $LysM\text{-}cre;Ctsd^{-/-}$ but significantly reduced in $MMTV\text{-}cre;Ctsd^{-/-}$ mice (Fig. 1a, b). We monitored tumor progression by palpation from 8 weeks on until the tumors reached an end-stage size of 1 cm$^3$. By the age of 12 weeks, half of the $LysM\text{-}cre;Ctsd^{-/-}$ and control mice had developed at least one palpable tumor, while this took 21 weeks for $MMTV\text{-}cre;Ctsd^{-/-}$ mice (Fig. 1c). At 18 weeks of age, we performed volumetric ultrasound imaging of the two largest tumors detectable in each individual mouse. Again, tumors from $MMTV\text{-}cre;Ctsd^{-/-}$ mice were significantly smaller than in control mice (Fig. 1d, e). Accordingly, we observed end-stage tumors occurring significantly later in $MMTV\text{-}cre;Ctsd^{-/-}$ mice (Fig. 1f). Half of the mice with a CTSD deletion in mammary epithelial cells reached an end-stage tumor burden at 28 weeks in contrast to $LysM\text{-}cre;Ctsd^{-/-}$ (20 weeks) and control (22 weeks) mice. In summary, deletion of CTSD in myeloid cells did not delay breast cancer progression, while CTSD deficiency in the mammary epithelium strongly impaired tumor development without affecting gross mammary gland function.

**Tumor cell-autonomous effects of CTSD deletion.** In order to study the role of CTSD in PyMT cancer cells, we generated a

tumor cell line from a $Ctsd^{fl/fl};mTmG$ mouse and introduced a doxycycline (Dox)-inducible cre recombinase expression system. In the presence of Dox, cells switch from a non-recombined (red fluorescent, CTSD-competent) to a recombined state (green fluorescent, CTSD-deficient). However, 17% of the cells already recombined in absence of Dox due to leakiness of the cre expression system (Fig. 2a). Nevertheless, the remaining 83% of the cells still produced reasonable amounts of CTSD protein (Fig. 2b, Day 0). Most importantly, after one day of Dox treatment, the majority of cells (typically 60 to 88% of the cells) recombined and showed reduced CTSD protein levels. Continued Dox treatment enriched for the recombined cells (>90%) and abrogated protein levels of the mature double-chain form of CTSD to not detectable by Western Blotting (Fig. 2a, b). We took advantage of obtaining a mixture of red CTSD-competent and green CTSD-deficient cells after a one-day Dox pulse and performed competitive growth assays in cell culture (Fig. 2c, d). In order to achieve a balanced number of recombined and non-recombined cells in these assays, we seeded cells and allowed them to grow for 3 days before inducing recombination by adding Dox for 24 h. The cell cycle is desynchronizing during the three-day culture period. As cre-mediated recombination occurs preferentially during S-phase[26], this regimen achieved a lower proportion of recombined cells (i.e., 60%; Fig. 2c, d) as compared to Dox application immediately after seeding (i.e., 84% in Fig. 2a). In these competitive cell growth experiments, recombined CTSD-deficient cells continuously decreased, while non-recombined CTSD-competent cells enriched (Fig. 2c). During tumor growth, cancer cells experience phases of nutrient restriction[27]. To mimic

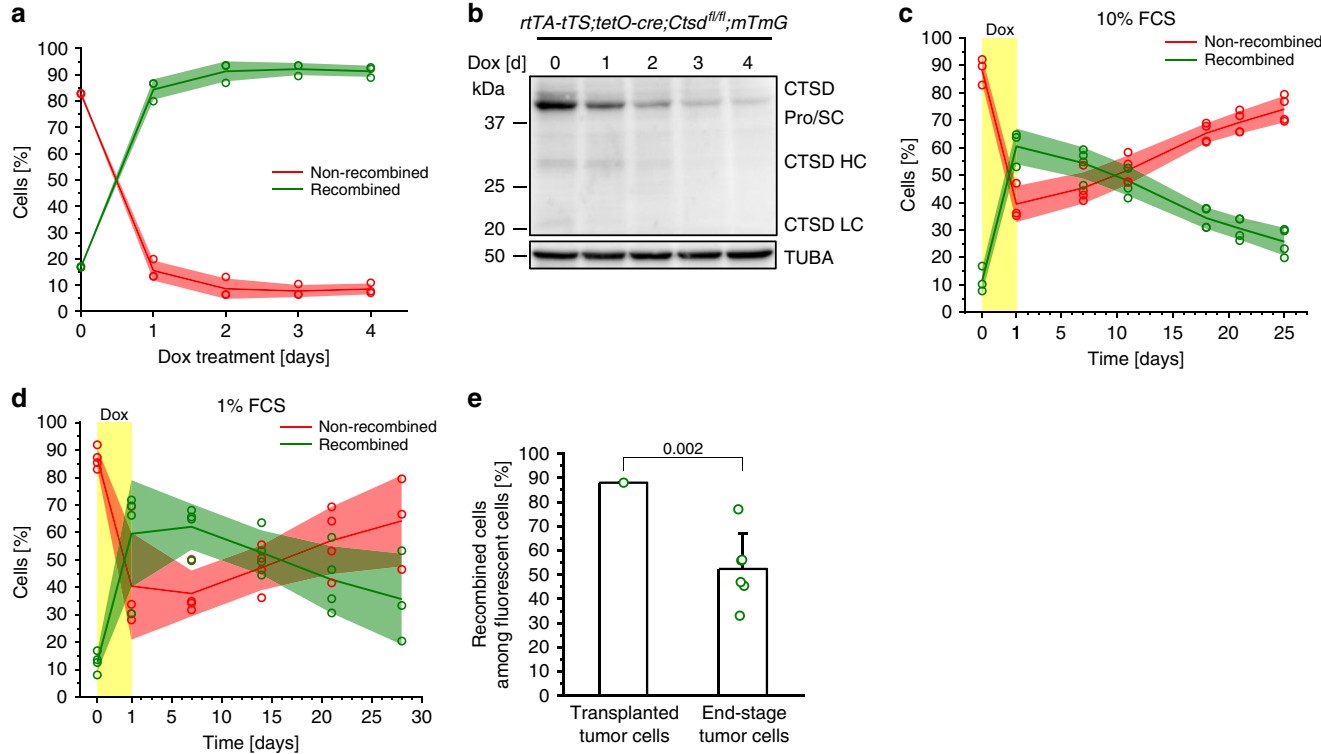

**Fig. 2 Tumor cell-autonomous growth deficit through CTSD deletion. a** Flow cytometry-based monitoring of cre recombination reporter expression (Tomato: non-recombined, GFP: recombined) in *rtTA-tTS;tetO-cre;Ctsd*^fl/fl^;*mTmG* PyMT cells treated for 0–4 days with doxycycline (Dox) (*n* = 3 independent experiments). **b** Analysis of CTSD expression by Western blot (with TUBA as loading control) in lysates from cells used in **a**. Pro/SC, zymogen/single-chain form; HC, heavy chain; LC, light chain. In vitro competitive growth assay of non-recombined and recombined *rtTA-tTS;tetO-cre; Ctsd*^fl/fl^;*mTmG* PyMT cells in 10% FCS (**c**) or 1% FCS (**d**) medium after a one-day pulse of Dox (*n* = 3 independent experiments for d 0 and d 1 in **c** and d 28 in **d**, *n* = 4 independent experiments for the rest). **e** In vivo competitive growth assay of non-recombined and recombined 10% FCS *rtTA-tTS;tetO-cre;Ctsd*^fl/ fl^;*mTmG* PyMT cells. A one-day Dox-pulsed cell suspension of known ratio of non-recombined to recombined cells (left bar) was orthotopically transplanted into 6 recipient mice. The ratio was again determined by flow cytometry in the outgrown tumors (right bar) (*n* = 6 animals; two-sided one-sample t-test). Line and bar charts show all data points with mean ± SD and *p*-value. Source data are provided as a Source Data file.

limited nutrient supply, we used a mild starvation medium, in which fetal calf serum (FCS) was reduced from 10 to 1%. Except for an initial lag-phase, we found again that CTSD-competent cells did outcompete the CTSD-deficient cells, albeit with more variation between the biological replicates (Fig. 2d). Growth of these cells was also tested in vivo by orthotopic transplantation into the mammary fat pad of *Ctsd* wild-type mice (Fig. 2e). In this experiment we injected a cell population containing 88% recombined CTSD-deficient cells independently into 6 recipient mice and analyzed the resulting tumors by flow cytometry detecting the mTmG-fluorescence. On average, the proportion of CTSD-deficient cells decreased to 52% in the outgrown tumors, while CTSD-competent cells rose from 12 to 48% of the tumor masses. Hence, CTSD-expressing PyMT cells have also a clear growth advantage in vivo. Together, these assays recapitulate the growth disadvantage of CTSD-deficient tumor cells seen in the primary *MMTV-cre;Ctsd*^−/−^ PyMT breast cancer model and provide evidence for tumor cell-autonomous functions of CTSD.

**CTSD deletion induces a quiescent cell state and expands the acidic cell compartment.** For mechanistic studies, we generated PyMT tumor cell lines from *Ctsd*^+/+^ and *MMTV-cre;Ctsd*^−/−^ tumor mice (Fig. 3a). A clear difference between the two cell lines manifested by culturing them in 1% FCS starvation medium. *Ctsd*^−/−^ cells accumulated many large intracellular vesicles, while there were barely any of these in *Ctsd*^+/+^ cells. Staining for acidic β-galactosidase activity was strongly positive in starved *Ctsd*^−/−^ cells (Fig. 3b). As high β-galactosidase activity is often used as a

marker for cellular senescence[28], we analyzed proliferation by a fluorescent label-retention assay. The number of label-retaining non-proliferating cells was higher in *Ctsd*^−/−^ cells compared to *Ctsd*^+/+^ cells, especially under starvation (Fig. 3c). Senescent cell states are also associated with increased sensitivity towards so-called senolytic agents. Culturing PyMT cells for 10 days in 1% FCS medium and subsequently treating them for 24 h with the senolytic agent Navitoclax, killed twice as many *Ctsd*^−/−^ cells as compared to *Ctsd*^+/+^ cells (Fig. 3d). As stalled proliferation and increased Navitoclax sensitivity suggested a senescent phenotype, we next explored the secretion of IL-6, a major indicator of a senescence-associated secretory phenotype (SASP). However, IL-6 secretion from CTSD-deficient cells was not increased, thereby excluding a SASP (Fig. 3e). To address negative regulators of cell cycle, quantitative RT-PCR for the cyclin-dependent kinase inhibitors Cdkn2a/p16 and Cdkn1a/p21 was performed. This showed a significant increase of *p16* mRNA in *Ctsd*^−/−^ cells, while *p21* expression was not affected (Fig. 3f). In summary, FCS-starved CTSD-deficient PyMT cells show an overall phenotype typical for quiescent cells as compared to the irreversible growth arrest occurring in bona fide cellular senescence[29]. The elevated activity of acidic β-galactosidase in *Ctsd*^−/−^ cells also suggests an increase in the acidic vesicular cell compartment. To test this further, flow cytometry using LysoTracker^TM^ Green was performed. The majority of *Ctsd*^+/+^ cells stained with medium intensity, both under normal and starving conditions, while *Ctsd*^−/−^ cells showed high LysoTracker^TM^ intensity, especially during starvation (Fig. 4a). A three-way ANOVA considering the

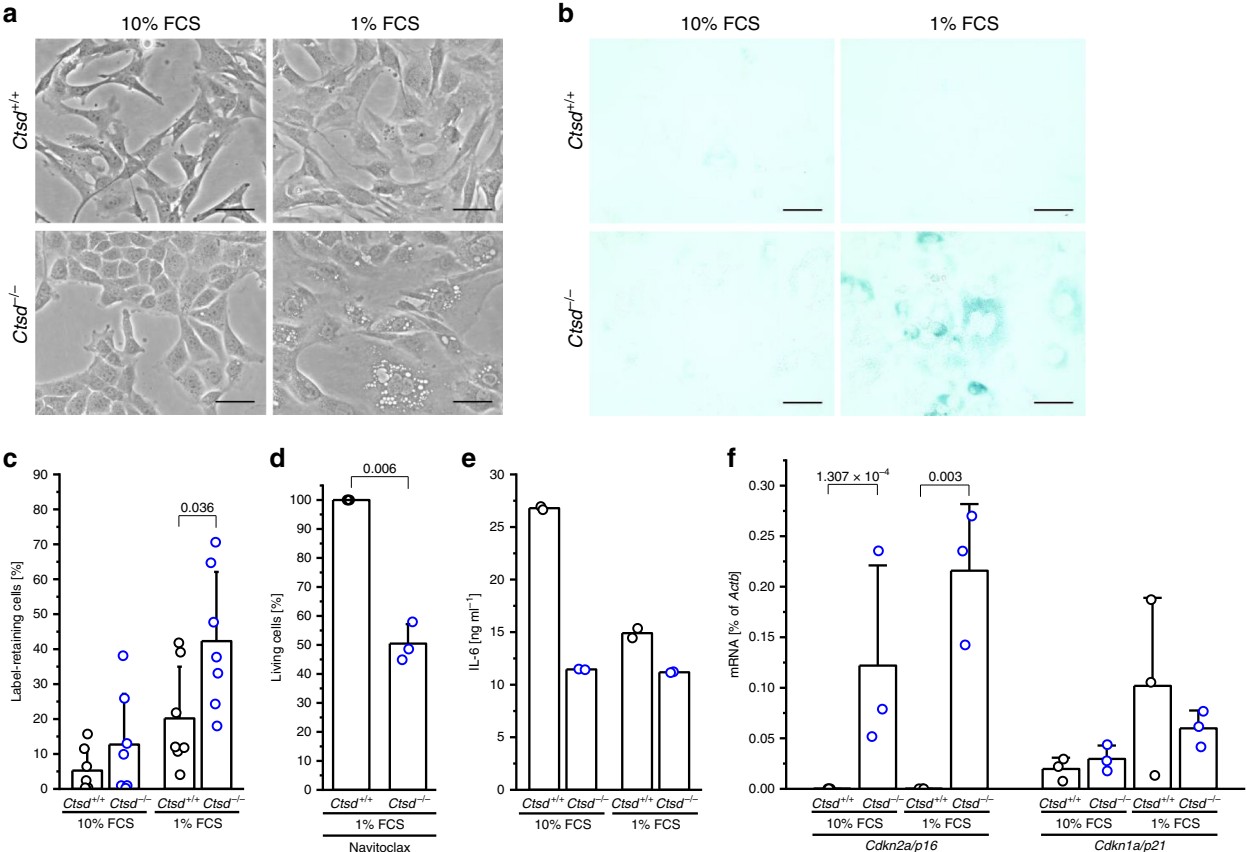

**Fig. 3 CTSD deficiency induces a quiescent cell state in short-term starved tumor cells. a** Pictures of $Ctsd^{+/+}$ and $Ctsd^{-/-}$ PyMT cells cultured for 7–10 days in 10% FCS or 1% FCS medium. Bars, 50 μm. Representative of 5 independent experiments. **b** Staining for acidic β-galactosidase in 10% FCS and 1% FCS $Ctsd^{+/+}$ and $Ctsd^{-/-}$ PyMT cells. Bars, 50 μm. Representative of 3 independent experiments. **c** Quantification of label-retaining cells among 10% FCS and 1% FCS $Ctsd^{+/+}$ and $Ctsd^{-/-}$ PyMT cells ($n = 7$ independent experiments; two-sided two-sample $t$-test). **d** Amount of Annexin-V⁻-7-AAD⁻(living) cells among 1% FCS $Ctsd^{+/+}$ and $Ctsd^{-/-}$ PyMT cells treated with Navitoclax, relative to DMSO control and $Ctsd^{+/+}$ ($n = 3$ independent experiments; two-sided one-sample $t$-test). **e** Quantification of IL-6 by alphaLISA in cell-conditioned media from 10% FCS and 1% FCS $Ctsd^{+/+}$ and $Ctsd^{-/-}$ PyMT cells ($n = 2$ independent experiments). **f** Relative $Cdkn2a/p16$ and $Cdkn1a/p21$ expression determined by RT-PCR in 10% FCS and 1% FCS $Ctsd^{+/+}$ and $Ctsd^{-/-}$ PyMT cells ($n = 3$ independent experiments; two-sided two-sample $t$-test). Bar charts show all data points with mean + SD and $p$-value. Source data are provided as a Source Data file.

factors intensity of LysoTracker™, $Ctsd$ genotype and FCS revealed significant dependence of the acidic cell compartment on $Ctsd$ ($p < 0.001$) and the amount of FCS in the culture medium ($p = 0.001$). The distribution of LysoTracker™ intensities in starved $Ctsd^{-/-}$ cells was highly reminiscent of the one of Torin-treated $Ctsd^{+/+}$ cells, the positive control for autophagy induction and lysosomal biogenesis[30]. Consistent with an increased acidic cell compartment, we found elevated levels of the lysosome-associated membrane protein 1 (LAMP1) when CTSD was missing (Fig. 4b). We also measured an upregulation of $Lamp1$ mRNA in $Ctsd^{-/-}$ cells, which was more pronounced under starving conditions (Fig. 4c). To assess whether the lysosomes that are being induced are functional, activities of lysosomal aspartic (CTSD, CTSE, and napsin) and cysteine proteases (including CTSB, CTSL, and CTSK) were measured by cleavage of fluorogenic peptides in lysates from normal and starved cells. As expected, aspartic protease activity was low in $Ctsd^{-/-}$ cells (Fig. 4d). In contrast to that, cysteine protease activity was not altered by CTSD deficiency, suggesting that absence of the major aspartic protease does not cause a general lack of lysosomal proteolysis (Fig. 4e). To test lysosomal proteolysis in intact PyMT cells, we loaded the acidic cell compartment with a quenched fluorogenic substrate by endocytosis. Degradation of this substrate by lysosomal proteases results in dequenching and

recovery of the fluorescent signal. Blocking the endocytic substrate uptake by Cytochalasin D or preventing lysosomal acidification by Bafilomycin A1 reduced lysosomal proteolysis (Fig. 4f). However, total lysosomal proteolysis was comparable between $Ctsd^{+/+}$ and $Ctsd^{-/-}$ cells, with an increase under FCS starvation. Interestingly, individual members of the same protease class responded differently to CTSD deficiency and/or FCS starvation. For example, CTSL protein was always increased in $Ctsd^{-/-}$ cells, but decreased in 1% FCS to a similar extent than in $Ctsd^{+/+}$ cells, despite an upregulation on mRNA level (Fig. 4g, h). In contrast to that, CTSB was rather unaffected by CTSD deficiency, but became induced under starvation, independent of the $Ctsd$ genotype (Fig. 4i). In summary, CTSD deficiency renders tumor cells quiescent, increases their lysosomal compartment under FCS starvation, and rewires the lysosomal proteolytic system.

**CTSD deficiency results in macroautophagy.** Cells cope with starvation by induction of macroautophagy, hereafter referred to as autophagy[31]. During autophagy, the cytosolic form of microtubule-associated protein 1 light chain 3 (LC3-I) is converted to LC3-II on the expanding phagophore and associates with p62, a cargo receptor targeting ubiquitylated proteins for autophagy. $Ctsd^{-/-}$ PyMT cells showed an increased

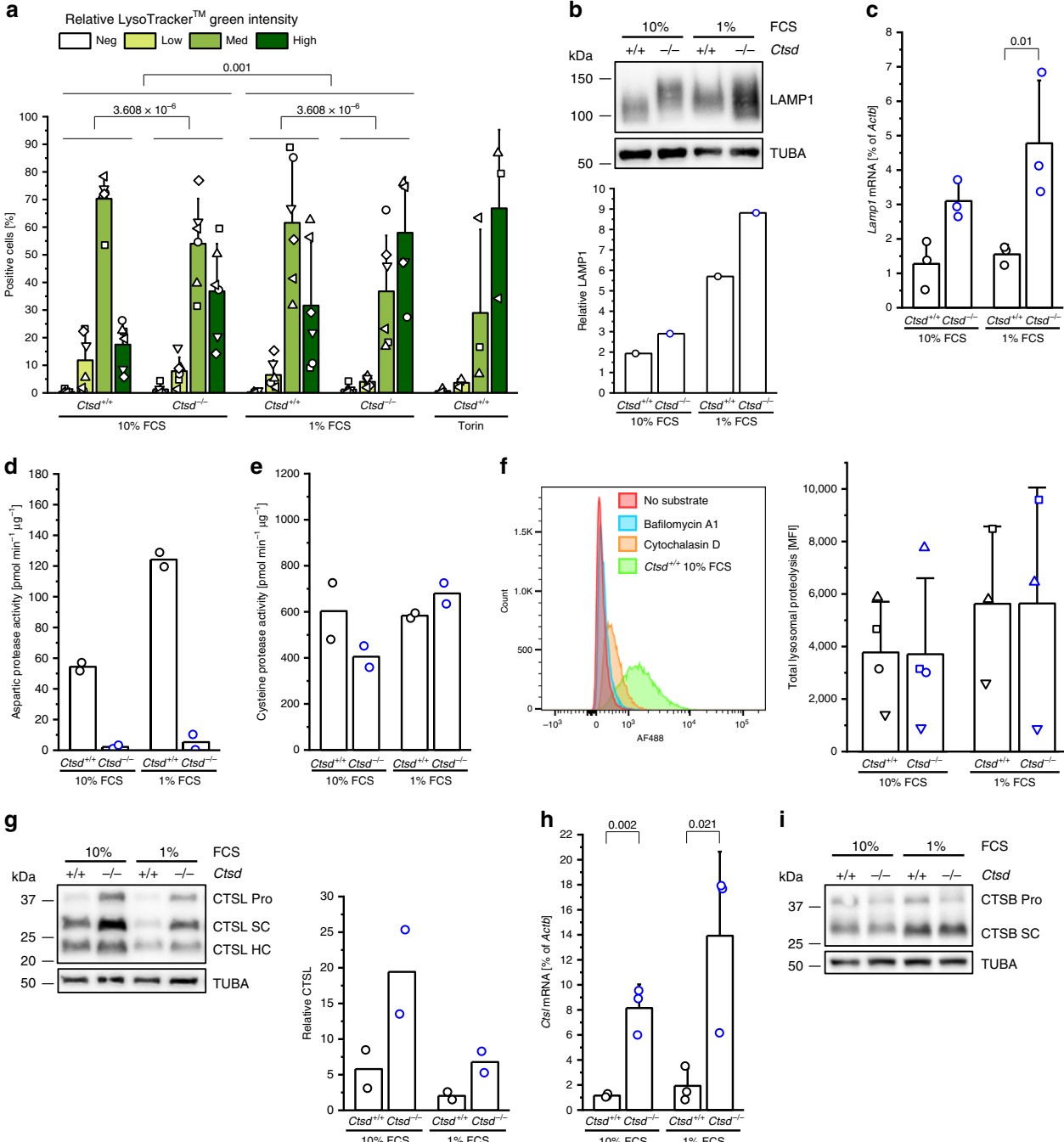

**Fig. 4 CTSD deficiency expands the acidic cell compartment of short-term starved tumor cells. a** Flow cytometry analysis of LysoTracker[TM] Green intensity in $Ctsd^{+/+}$ and $Ctsd^{-/-}$ PyMT cells cultured for 7–10 days in 10% FCS or 1% FCS medium or treated with Torin ($n = 3$ independent experiments for Torin, $n = 6$ independent experiments for the rest; two-sided three-way ANOVA for the factors intensity of LysoTracker[TM], $Ctsd$ genotype and FCS). **b** Analysis of LAMP1 expression by Western blot (with TUBA as loading control) in 10% FCS and 1% FCS $Ctsd^{+/+}$ and $Ctsd^{-/-}$ PyMT cells. Slight molecular weight changes due to different glycosylation pattern. Representative of 2 independent experiments. Quantification is below ($n = 1$ experiment). **c** Relative $Lamp1$ expression determined by RT-PCR in 10% FCS and 1% FCS $Ctsd^{+/+}$ and $Ctsd^{-/-}$ PyMT cells ($n = 3$ independent experiments; two-sided two-sample $t$-test). Activity assay with substrates specific for aspartic (**d**) or cysteine (**e**) proteases in 10% FCS and 1% FCS $Ctsd^{+/+}$ and $Ctsd^{-/-}$ PyMT cells ($n = 2$ independent experiments). **f** Lysosomal activity assay using a quenched fluorogenic substrate in 10% FCS and 1% FCS $Ctsd^{+/+}$ and $Ctsd^{-/-}$ PyMT cells ($n = 4$ independent experiments for 10% FCS, $n = 3$ independent experiments for 1% FCS). Histogram including controls blocking the endocytic substrate uptake (Cytochalasin D) or lysosomal acidification (Bafilomycin A1) on the left, median fluorescence intensity (MFI) on the right. **g** Analysis of CTSL expression by Western blot (with TUBA as loading control) in 10% FCS and 1% FCS $Ctsd^{+/+}$ and $Ctsd^{-/-}$ PyMT cells. Pro, zymogen; SC, single-chain form; HC, heavy chain. Representative of 2 independent experiments. Quantification is to the right ($n = 2$ independent experiments). **h** Relative $Ctsl$ expression determined by RT-PCR in 10% FCS and 1% FCS $Ctsd^{+/+}$ and $Ctsd^{-/-}$ PyMT cells ($n = 3$ independent experiments; two-sided two-sample $t$-test). **i** Analysis of CTSB expression by Western blot (with TUBA as loading control) in 10% FCS and 1% FCS $Ctsd^{+/+}$ and $Ctsd^{-/-}$ PyMT cells. Pro, zymogen; SC, single-chain form. Representative of 2 independent experiments. Bar charts show all data points with mean + SD and $p$-value. Source data are provided as a Source Data file.

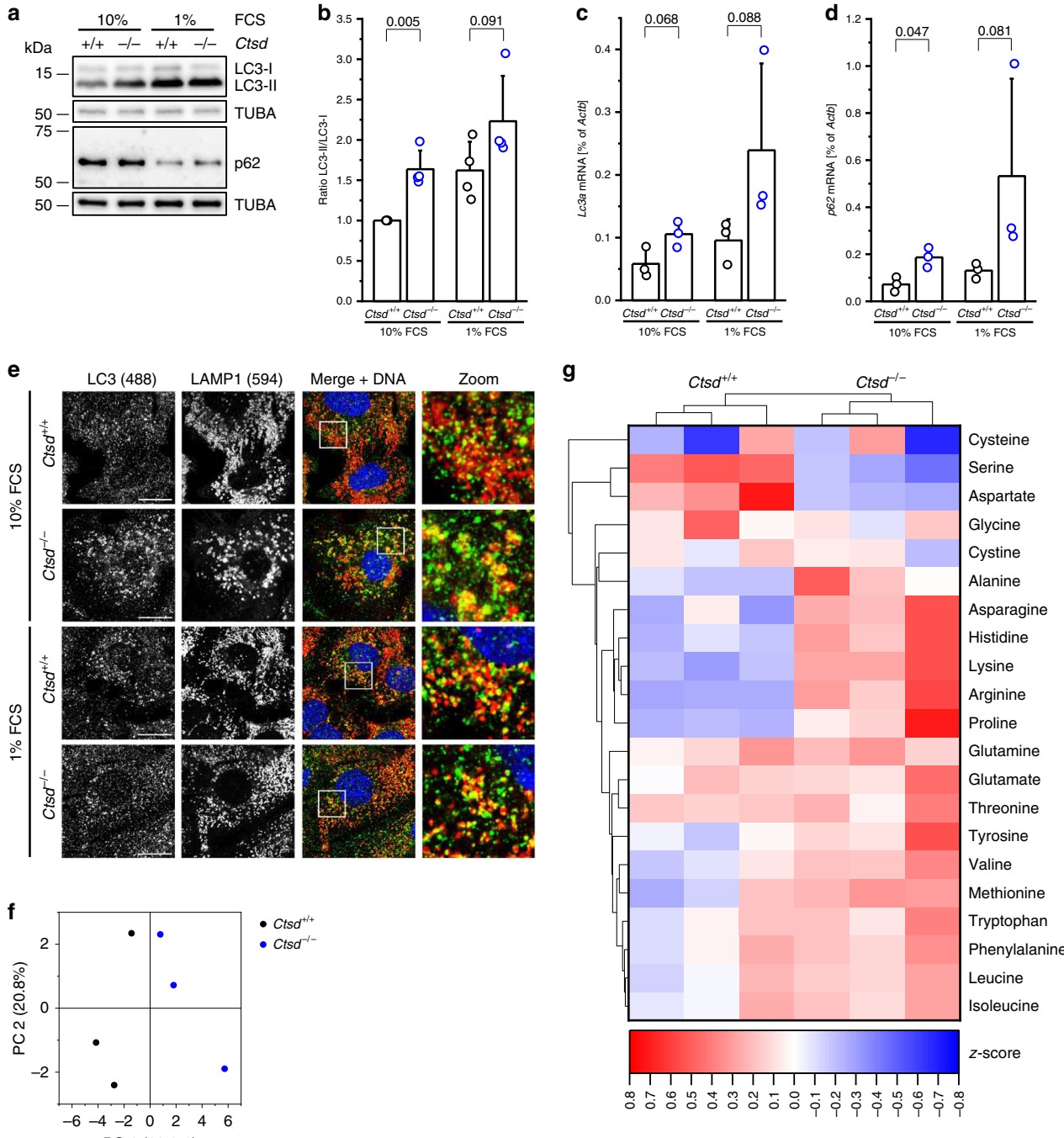

**Fig. 5 CTSD deficiency induces autophagy in short-term starved tumor cells. a** Analysis of LC3-I, LC3-II, and p62 protein levels by Western blot (with TUBA as loading control) in $Ctsd^{+/+}$ and $Ctsd^{-/-}$ PyMT cells cultured for 7–10 days in 10% FCS or 1% FCS medium. Representative of 4 independent experiments. **b** Quantification of LC3 Western blots as in **a**, plotted as LC3-II/LC3-I ratio relative to 10% FCS $Ctsd^{+/+}$ ($n = 4$ independent experiments; two-sided one-sample and two-sample $t$-test). Relative $Lc3a$ (**c**) and $p62$ (**d**) expression determined by RT-PCR in 10% FCS and 1% FCS $Ctsd^{+/+}$ and $Ctsd^{-/-}$ PyMT cells ($n = 3$ independent experiments; two-sided two-sample $t$-test). **e** Pictures of 10% FCS and 1% FCS $Ctsd^{+/+}$ and $Ctsd^{-/-}$ PyMT cells stained for LC3 (green), LAMP1 (red), and DNA (blue). Zoom shows enlargement of indicated image area in merge+DNA. Bars, 10 μm. Representative of 3 independent experiments. **f** Principle component analysis (PCA) of targeted metabolomics data from 1% FCS $Ctsd^{+/+}$ and $Ctsd^{-/-}$ PyMT cells ($n = 3$ independent experiments). **g** Clustering analysis of the relative abundance of all 20 canonical amino acids and cystine in 1% FCS $Ctsd^{+/+}$ and $Ctsd^{-/-}$ PyMT cells shown as heatmap ($n = 3$ independent experiments). Bar charts show all data points with mean + SD and $p$-value. Source data are provided as a Source Data file.

LC3-II/LC3-I ratio as well as an induction of $Lc3a$ mRNA expression when cultured in presence of 10% FCS (Fig. 5a–c). These differences did not reach significance in FCS-starved medium, most likely because of the induction of autophagy in $Ctsd^{+/+}$ control cells. Together these data suggest that autophagy

is induced and also successfully executed in $Ctsd^{-/-}$ cells. As a further indicator for that, protein levels of the autophagy cargo receptor p62 were reduced upon FCS starvation in $Ctsd^{+/+}$ and $Ctsd^{-/-}$ cells, probably due to p62 degradation in the autolyso-some (Fig. 5a). Of note, the $Ctsd$ genotype did not affect p62

protein levels, despite an upregulation of *p62* mRNA in *Ctsd*$^{-/-}$ cells (Fig. 5d). In order to further assess autophagic flux, cells in FCS- and amino acid-free medium were treated with Bafilomycin A1 or a combination of Pepstatin A and E64d to block lysosomal acidification and protease activity, respectively (Supplementary Fig. 2). Both treatments led to an increase of the LC3-II/LC3-I ratio followed by the accumulation of p62 protein, independent of the *Ctsd* genotype. This clearly indicates that the proteolytic degradation of autolysosomal content in *Ctsd*$^{-/-}$ cells is intact, which is also supported by the maintained global proteolytic capacity of CTSD-deficient PyMT cells reported in Fig. 4f. Consequently, the increased LC3-II/LC3-I ratio seen in *Ctsd*$^{-/-}$ cells under FCS starvation is not a result of perturbed degradation but represents an increase of autophagic flux. Furthermore, immunofluorescent co-staining of autophagosomal LC3 and lysosomal LAMP1 showed equal co-localization in *Ctsd*$^{+/+}$ and *Ctsd*$^{-/-}$ cells under normal and starving conditions (Fig. 5e). In synopsis, the data suggest an unperturbed formation of autolysosomes and an increased autophagic flux with the generation of free amino acids in *Ctsd*$^{-/-}$ cancer cells. To test this, we performed a targeted metabolomics approach to analyze the 20 canonical amino acids and cystine in starved *Ctsd*$^{+/+}$ and *Ctsd*$^{-/-}$ cells. The principal component analysis (PCA) of the data revealed a clear separation of *Ctsd*$^{+/+}$ and *Ctsd*$^{-/-}$ cells (Fig. 5f). CTSD-expressing cells showed a relatively balanced amino acid composition with average z-scores being positive for 10 amino acids and negative for 11 amino acids (Fig. 5g). *Ctsd*$^{-/-}$ cells differed from *Ctsd*$^{+/+}$ cells in that the majority of amino acids (16 of 21) had a positive average z-score (Fig. 5g). These data indicate that CTSD deficiency does not cause a general block of cellular protein degradation for the generation of free amino acids and are in line with an increased activity of macroautophagy in *Ctsd*$^{-/-}$ breast cancer cells.

Next, we complemented the cell culture studies by investigation of primary cancers from *MMTV-cre;Ctsd*$^{-/-}$ and control PyMT mice. The transcriptome of tumors from 18-week-old mice obtained by RNA sequencing (RNA-Seq) showed a significant number of genes important for lysosomal biogenesis and function being upregulated in *MMTV-cre;Ctsd*$^{-/-}$ tumors compared to controls (Supplementary Fig. 3a). A gene set enrichment analysis (GSEA) revealed that among the autophagy-related GO terms eight were significantly upregulated ($q < 0.05$) (Supplementary Fig. 3b). These terms included macroautophagy but not selective autophagy processes such as chaperone-mediated autophagy. Thus, these analyses of primary tumors also provide evidence for an upregulation of lysosomal biogenesis and autophagy during CTSD deficiency.

**Deregulated mTORC1 signaling in CTSD-deficient breast cancer cells**. To link the proliferation block and autophagy induction with the deficiency of a lysosomal protease and starvation, we set out to investigate molecular pathways regulating lysosomal biogenesis and autophagy. Phosphorylation of p38 MAPK has been shown to upregulate the transcription of lysosome and autophagy genes through inhibition of the transcriptional repressor ZKSCAN3[32]. FCS starvation activated the stress-responsive p38 MAPK irrespective of the *Ctsd* genotype (Fig. 6a). However, in both FCS conditions, phosphorylation of p38 MAPK was markedly reduced in *Ctsd*$^{-/-}$ cells as compared to *Ctsd*$^{+/+}$ cells. Because these results cannot completely explain the increased autophagy of FCS-starved *Ctsd*$^{-/-}$ cells, we next examined the mTORC1 signaling pathway. mTORC1 controls anabolic and catabolic processes in response to energy and nutrient levels[33]. In presence of growth factors and amino acids, mTORC1 is known to be recruited to the lysosomal surface where it is active. Subsequent phosphorylation of the 70 kDa ribosomal

Protein S6 Kinase B1 (P70S6K), with ribosomal protein S6 being its main target, is crucial for promoting cell growth. *Ctsd*$^{-/-}$ cells exhibited lower levels of phosphorylated P70S6K and S6 both under normal and starving conditions, which fits the observed phenotype of *Ctsd*$^{-/-}$ cells (Fig. 6b, c). Therefore, we wondered if the impaired mTORC1 signaling might be attributed to a mislocalization of mTORC1. We performed confocal immunofluorescent co-staining of mTOR and LAMP1 and determined the degree of co-localization by the Pearson correlation coefficient *r* (Fig. 6d–g). mTOR showed a diffuse cytosolic staining pattern, and lysosomes were more abundant in normal and amino acid-starved *Ctsd*$^{-/-}$ cells. Co-localization of mTOR and LAMP1 in *Ctsd*$^{+/+}$ cells was reduced in amino acid-starved conditions when compared to complete medium (10% FCS: $r = 0.406$; w/o FCS, AA: $r = 0.241$). Importantly, mTOR/LAMP1 co-localization was essentially lost in *Ctsd*$^{-/-}$ cells (10% FCS: $r = 0.081$; w/o FCS, AA: $r = 0.056$). We were able to rescue mTOR/LAMP1 co-localization in amino acid-starved *Ctsd*$^{+/+}$ cells by the addition of amino acids, but failed to do so in *Ctsd*$^{-/-}$ cells (Fig. 6f, g). Notably, treating amino acid-starved *Ctsd*$^{+/+}$ cells with the aspartic protease inhibitor Pepstatin A resulted in a displacement of mTOR from the lysosome to a higher extent as compared to amino acid starvation alone (Supplementary Fig. 4a, b).

Next, we aimed to address effects of CTSD deficiency on factors known to tether mTOR to the lysosome and to investigate mTORC1 assembly. Lysosomal localization of mTOR has been shown to depend on RagGTPases that are recruited to the lysosome by a pentameric complex called Ragulator[33]. In response to sufficient amino acid levels the RagGTPases bind the mTORC1 component Raptor, thereby placing mTOR at the lysosomal surface. First, we investigated protein levels of the two interaction partners Raptor and RagC. Comparing *Ctsd*$^{-/-}$ to *Ctsd*$^{+/+}$ cells, there was no lack of any of the two proteins (Supplementary Fig. 5a). Second, we assessed the assembly of mTORC1. To this end, the mTORC1 component mLST8 was overexpressed as an EGFP-fusion protein in *Ctsd*$^{+/+}$ and *Ctsd*$^{-/-}$ cells. Using GFP-Trap beads, we pulled down the EGFP-fusion protein and its interaction partners. EGFP-mLST8 pull-downs showed immunoreactivity with mTOR and Raptor antibodies, while the EGFP-only control showed no signal (Supplementary Fig. 5b). The EGFP-mLST8 pull-down was successful in both *Ctsd* genotypes, indicating a correct mTORC1 assembly in *Ctsd*$^{-/-}$ cells. Next, mTORC1-interacting proteins at the lysosomal membrane were investigated for their expression levels and localization in *Ctsd*$^{+/+}$ and *Ctsd*$^{-/-}$ cells. Protein levels of Rheb, the small GTPase important for growth factor signal transmission and mTOR activation, as well as the Ragulator component Lamtor4 were equally present (Supplementary Fig. 5c). Furthermore, LAMTOR4 co-localization with LAMP1 did not differ between *Ctsd*$^{+/+}$ and *Ctsd*$^{-/-}$ cells (Supplementary Fig. 5d).

Together, these data indicate that lysosomes of CTSD-deficient cells are equipped with important mTORC1 interaction partners and principal mTORC1 assembly is not impaired by CTSD deficiency. However, mTORC1 association at the lysosomal surface is substantially impaired in absence of CTSD. This suggests that aberrant lysosomes may be the basis for the sustained displacement of mTOR from lysosomes in *Ctsd*$^{-/-}$ cells.

**CTSD-deficient breast cancers escape the growth blockade with a two-month delay**. Tumorigenesis in *MMTV-cre;Ctsd*$^{-/-}$ mice is delayed by about two months (Fig. 1c). However, once tumors had developed in *MMTV-cre;Ctsd*$^{-/-}$ mice, they progressed to end-stage as fast as in control mice (Fig. 7a). In addition, the *Ctsd* genotype did not affect the metastatic burden in the lungs of end-stage cancer

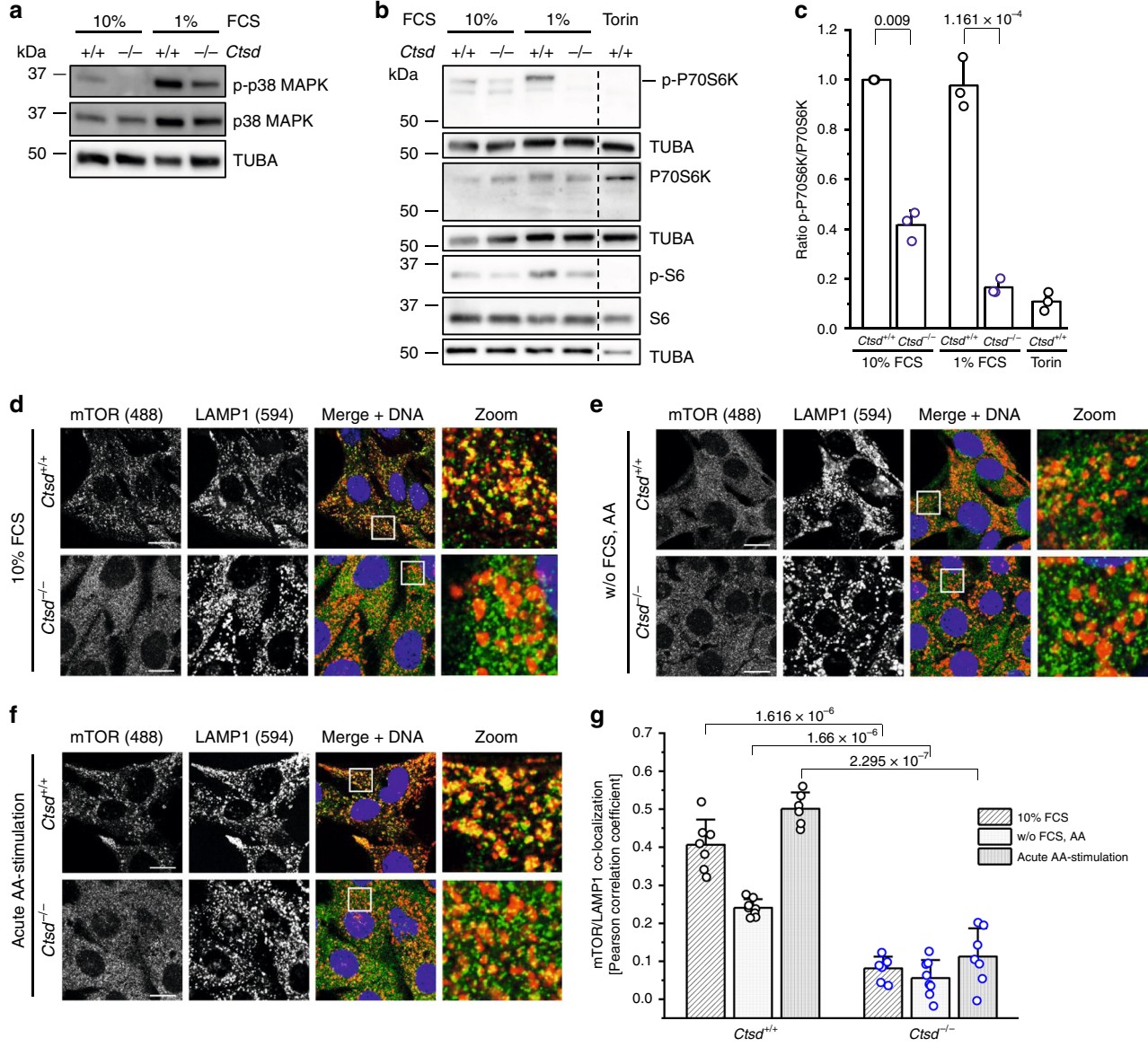

**Fig. 6 CTSD deficiency impairs mTORC1 localization and activity at the lysosomal membrane. a** Analysis of phosphorylated (p-p38 MAPK) and total p38 MAPK protein levels by Western blot (with TUBA as loading control) in $Ctsd^{+/+}$ and $Ctsd^{-/-}$ PyMT cells cultured for 7–10 days in 10% FCS or 1% FCS medium. Representative of 2 (p-p38 MAPK) and 1 (p38 MAPK) independent experiments. **b** Analysis of phosphorylated (p-P70S6K and p-S6) and total P70S6K and S6 protein levels by Western blot (with TUBA as loading control) in 10% FCS, 1% FCS, and mTOR inhibitor Torin-treated $Ctsd^{+/+}$ and $Ctsd^{-/-}$ PyMT cells. Representative of 2 ((p-)S6) and 3 ((p-)P70S6K) independent experiments. **c** Quantification of (p-)P70S6K Western blots as in **b**, plotted as p-P70S6K/P70S6K ratio relative to 10% FCS $Ctsd^{+/+}$ ($n = 3$ independent experiments; two-sided one-sample and two-sample $t$-test). Pictures of 10% FCS (**d**), FCS- and amino acid (AA)-starved (**e**), and acutely AA-stimulated (**f**) $Ctsd^{+/+}$ and $Ctsd^{-/-}$ PyMT cells stained for mTOR (green), LAMP1 (red), and DNA (blue). Zoom shows enlargement of indicated image area in merge+DNA. Bars, 10 µm. Representative of 2 independent experiments. **g** Quantification of mTOR/LAMP1 co-localization in **d–f** ($n = 6$ images for acutely AA-stimulated $Ctsd^{+/+}$, $n = 7$ images for acutely AA-stimulated $Ctsd^{-/-}$ and 10% FCS, $n = 8$ images for w/o FCS, AA; two independent experiments; two-sided two-sample $t$-test). Bar charts show all data points with mean + SD and $p$-value. Source data are provided as a Source Data file.

mice (Fig. 7b). Apparently, cancers found a way to escape the growth arrest imposed by CTSD deficiency after a two-month latency period—just to grow and metastasize seemingly undisturbed. Of note, also end-stage $MMTV$-$cre;Ctsd^{-/-}$ tumors were CTSD-deficient in their cancer cell compartment (Supplementary Fig. 1b). We hypothesized that $MMTV$-$cre;Ctsd^{-/-}$ tumors may have acquired oncogenic mutations during the long latency of tumor development. Therefore, we sequenced the whole exome of tumors from 18-week-old mice (Fig. 7c, d). We identified single nucleotide variants (SNV) and small insertions or deletions (Indel) that arose through somatic mutations or loss of

heterozygosity (LOH). In general, all tumors exhibited low numbers of mutations, which confirms previous reports showing highest genomic stability of PyMT tumors among all investigated breast cancer models[34]. Comparing the mutation types of $MMTV$-$cre;Ctsd^{-/-}$ and control tumors, we did not observe major differences in frequency, except for intergenic mutations (23 vs 11%) and the absence of splice region, non-coding, and stop mutations in $MMTV$-$cre;Ctsd^{-/-}$ tumors (Fig. 7c). Circos plots showed that the majority of mutations were unique for one individual and that tumors from $MMTV$-$cre;Ctsd^{-/-}$ mice on average did not have more mutations than the controls (Fig. 7d).

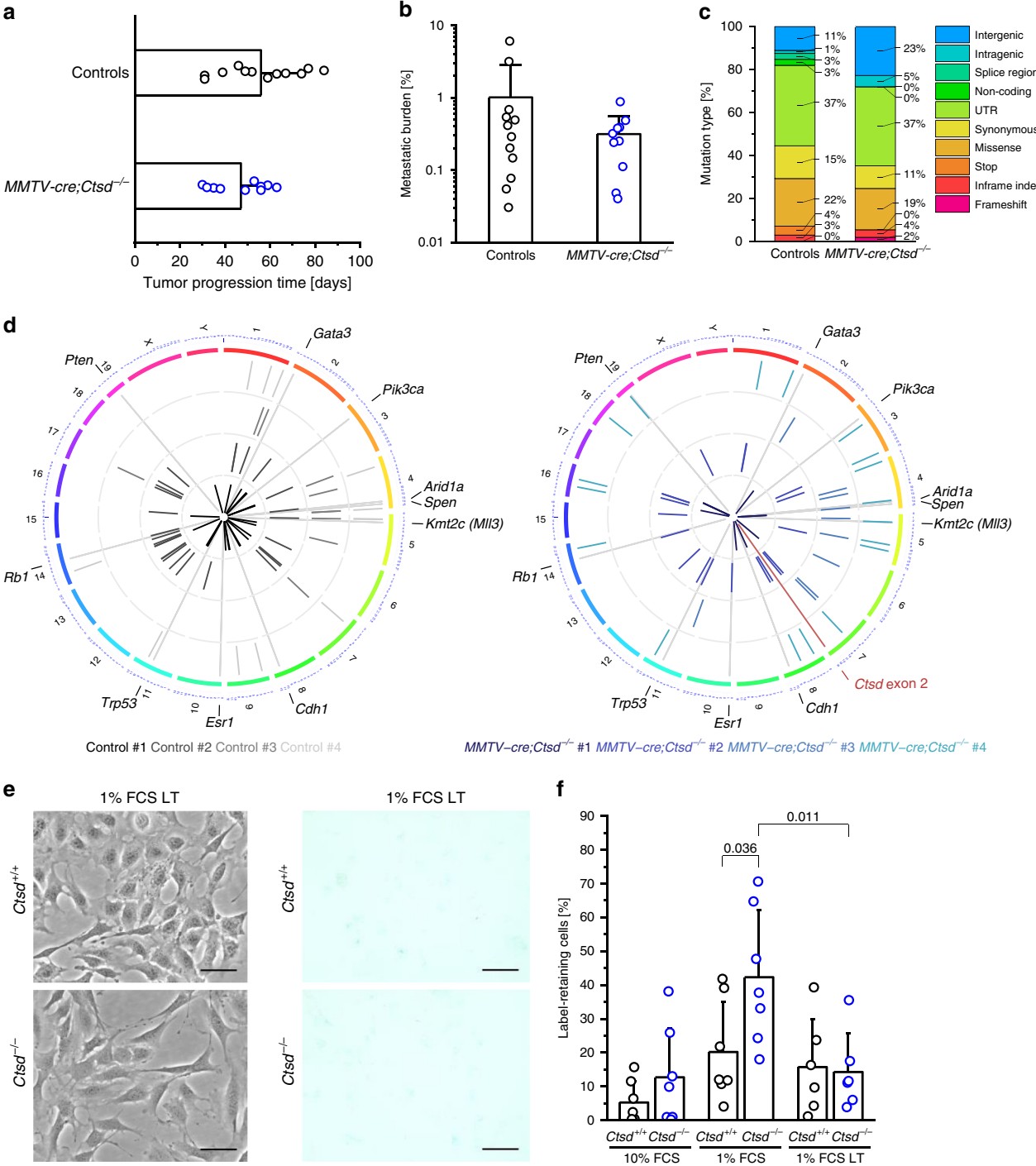

**Fig. 7 CTSD-deficient tumors overcome the initial growth arrest independent of somatic mutations. a** Time from first palpation to end-stage for tumors of PyMT mice of indicated genotypes ($n = 12$ animals for controls ($4\times Ctsd^{+/fl} + 8\times Ctsd^{fl/fl}$), $n = 10$ animals for *MMTV-cre;Ctsd*$^{-/-}$). **b** Metastatic burden in lungs from PyMT mice of indicated genotypes with end-stage tumors ($n = 12$ animals for controls ($4\times Ctsd^{+/fl} + 8\times Ctsd^{fl/fl}$), $n = 10$ animals for *MMTV-cre;Ctsd*$^{-/-}$). **c** Proportion of different types of somatic and loss of heterozygosity (LOH) mutations detected by whole exome sequencing in tumors from 18-week-old PyMT mice of indicated genotypes (Controls: *Ctsd*$^{fl/fl}$; $n = 4$ animals). **d** Circos plots showing mutations from **c** along the chromosomes per individual control (left) and *MMTV-cre;Ctsd*$^{-/-}$ (right) tumor (represented as separate concentric rings). Deletion of *Ctsd* exon 2 is highlighted in red. Continuous gray lines mark positions of homologs of frequently mutated genes in human breast cancer. **e** Pictures of unstained (left) and acidic β-galactosidase-stained (right) *Ctsd*$^{+/+}$ (top) and *Ctsd*$^{-/-}$ (bottom) PyMT cells cultured for ≥8 weeks in 1% FCS medium (1% FCS LT). Compare to Fig. 3a, b for short-term 1% FCS *Ctsd*$^{+/+}$ and *Ctsd*$^{-/-}$ PyMT cells. Bars, 50 μm. Representative of 2 independent experiments. **f** Quantification of label-retaining cells among 1% FCS LT *Ctsd*$^{+/+}$ and *Ctsd*$^{-/-}$ PyMT cells ($n = 6$ independent experiments; two-sided two-sample *t*-test). Short-term conditions (10% FCS, 1% FCS) as presented in Fig. 3c are included for convenient comparison. Bar charts show all data points with mean + SD and *p*-value. Source data are provided as a Source Data file.

The only consistent genetic alteration we found in tumors of *MMTV-cre;Ctsd*$^{-/-}$ mice was the deletion of the intentionally targeted exon 2 of the *Ctsd* gene. Furthermore, none of the observed mutations mapped to the 10 most frequently mutated genes in human breast cancer. To conclude, there was no accumulation of DNA alterations that could explain the eventual tumor outgrowth in *MMTV-cre;Ctsd*$^{-/-}$ PyMT mice.

To further study the escape of the transient growth block caused by CTSD deficiency, we extended the cultivation time of tumor cells in 1% FCS starvation medium from < 2 weeks to ≥ 8 weeks, a condition we defined as 1% FCS long-term (1% FCS LT). Remarkably, 1% FCS LT *Ctsd*$^{-/-}$ tumor cells lost their β-galactosidase-positive vesicles (Fig. 7e; compare to Fig. 3b) and numbers of non-proliferating cells were not different from 1% FCS LT *Ctsd*$^{+/+}$ cells (Fig. 7f). The increase of lysosomal LAMP1 protein seen in short-term starved CTSD-deficient cells declined upon LT starvation (Supplementary Fig. 6a). Similarly, the LC3-II/LC3-I ratio as well as p62 protein and mRNA returned to levels as found in 10% FCS (Supplementary Fig. 6b–d), indicating that macroautophagy returned to its basic flux state. In stark contrast to this reversion of the quiescent cell state, phosphorylation of P70S6K and S6 in 1% FCS LT *Ctsd*$^{-/-}$ was as low as in 1% FCS short-term *Ctsd*$^{-/-}$ cells and mTOR co-localization with LAMP1 was still significantly reduced in 1% FCS LT *Ctsd*$^{-/-}$ cells (Supplementary Fig. 6e–h). Thus, mTORC1 signaling continues to be perturbed in 1% FCS LT *Ctsd*$^{-/-}$ cells and resumption of proliferation must occur by different mechanisms.

**CTSD-deficient cells rewire oncogenic cellular signaling and gene expression upon long-term FCS starvation.** To address the rescue mechanisms in long-term starved *Ctsd*$^{-/-}$ cells, we performed RNA-Seq transcriptome analysis with short- and long-term starved *Ctsd*$^{+/+}$ and *Ctsd*$^{-/-}$ cells. Of the 12483 quantified transcripts, we identified the expressed genes with a significant differential regulation of at least two ($|\log2$ fold change (FC)$| > 1$; adjusted *p*-value $< 0.05$). The 1066 downregulated and the 1429 upregulated genes were subjected to STRING, the Search Tool for the Retrieval of Interacting Genes/Proteins[35], for unsupervised clustering (Supplementary Fig. 7, Supplementary Fig. 8). Within the downregulated genes in 1% FCS LT *Ctsd*$^{-/-}$ cells we found small clusters of keratins (I), desmosome components (II) and tight junction proteins (III), and larger clusters of G proteins and G Protein Coupled Receptors (GPCR), Rho GTPase-Activating Proteins (GAP) and Guanine nucleotide Exchange Factors (GEF), proteins for trafficking and endocytosis as well as ubiquitin ligases (Supplementary Fig. 7). While the aforementioned large clusters also occurred within the upregulated genes, the clusters describing epithelial cell-cell contacts were specific for the downregulated transcripts. Clusters uniquely found within the upregulated genes were mitosis (Supplementary Fig. 8: violet nodes in I), proteins involved in Transforming Growth Factor beta (TGFβ)/ Bone Morphogenetic Protein (BMP) signaling (Supplementary Fig. 8: red nodes in II) and epithelial-to-mesenchymal transition (EMT) (Supplementary Fig. 8: blue nodes in II), collagens (Supplementary Fig. 8: III), and A Disintegrin and Metalloproteinase with Thrombospondin motifs (ADAMTS) metalloproteases (Supplementary Fig. 8: IV). A GSEA yielded 42 significantly regulated KEGG (Kyoto Encyclopedia of Genes and Genomes) pathways in 1% FCS LT *Ctsd*$^{-/-}$ cells (Table 1). The top upregulated pathways were mainly connected to oncogenic signaling, including the Ras-ERK and PI3K-Akt pathways. In summary of the RNA-Seq, *Ctsd*$^{-/-}$ cells lose epithelial cell-cell contacts and acquire mesenchymal traits as well as a more pronounced oncogenic gene expression profile upon long-term starvation. Indeed, 1% FCS LT *Ctsd*$^{-/-}$ cells appeared less epithelial as compared to 1% FCS

**Table 1 Re-growing CTSD-deficient tumor cells upregulate oncogenic signaling pathways.**

| Downregulated | FDR |
|---|---|
| Tight junction | 1.90E−04 |
| Arginine and proline metabolism | 1.25E−02 |
| **Upregulated** | **FDR** |
| Pathways in cancer | 3.75E−06 |
| Rap1 signaling pathway | 1.18E−05 |
| Focal adhesion | 1.51E−05 |
| PI3K-Akt signaling pathway | 2.65E−05 |
| ECM-receptor interaction | 1.30E−04 |
| AGE-RAGE signaling pathway in diabetic complications | 5.20E−04 |
| Axon guidance | 5.70E−04 |
| Relaxin signaling pathway | 6.30E−04 |
| MAPK signaling pathway | 7.90E−04 |
| Ras signaling pathway | 1.40E−03 |

Search Tool for the Retrieval of Interacting Genes/Proteins (STRING) -based KEGG pathway enrichment analysis on transcripts expressed and significantly regulated (1066 down, 1429 up) in *Ctsd*$^{-/-}$ PyMT cells cultured for ≥8 weeks in 1% FCS medium (1% FCS LT) compared to 1% FCS *Ctsd*$^{-/-}$ PyMT cells and corrected for long-term 1% FCS culture effects in *Ctsd*$^{+/+}$ PyMT cells ($n = 3$). Top 10 pathways with false discovery rate (FDR).

short-term *Ctsd*$^{-/-}$ cells (compare Figs. 7e and 3a). The acquisition of a mesenchymal cell morphology in *Ctsd*$^{-/-}$ cells following long-term starvation was accompanied by a significant increase of the mesenchymal markers N-Cadherin and Vimentin both on mRNA and protein level (Supplementary Fig. 9a-c). The rescue of proliferation seen in 1% FCS LT *Ctsd*$^{-/-}$ was also evident on transcriptome level, but could not be attributed to a single KEGG signaling pathway (Table 1).

As kinase activity is essential for cellular signaling, we evaluated the phosphorylation status of kinase substrates by antibody arrays. To pinpoint pathways explaining the rescue of *Ctsd*$^{-/-}$ cell proliferation in long-term cultures, we searched for proteins that were differentially phosphorylated in 1% FCS LT *Ctsd*$^{-/-}$ vs 1% FCS short-term *Ctsd*$^{-/-}$ cells and in 1% FCS LT *Ctsd*$^{-/-}$ vs 1% FCS LT *Ctsd*$^{+/+}$ cells, but not in *Ctsd*$^{-/-}$ vs *Ctsd*$^{+/+}$ cells kept for short-term in 1% FCS. This was exclusively true for CREB, the cAMP-Responsive Element Binding protein (Fig. 8a). CREB phosphorylation on Ser[133] activates transcription of its target genes, which have been compiled in a specific gene set[36]. Making use of our RNA-Seq data, we plotted the log2 FC of significantly regulated genes of the CREB gene set to compare all conditions and to identify induced genes (log2 FC > 1). Indeed, comparing 1% FCS long- and short-term *Ctsd*$^{-/-}$ culture showed the most induced CREB target genes (Fig. 8b). Next, we compared the change in expression of these 32 genes between the different culture conditions (Fig. 8c). While being mostly downregulated in *Ctsd*$^{-/-}$ cells under short-term starvation, they became specifically upregulated in long-term starved *Ctsd*$^{-/-}$ cells. These genes regulate the actin cytoskeleton, cellular signaling, mitosis, and DNA repair. Thus, it is not surprising that many of them have been reported to be overexpressed and/or tumor-promoting in cancer. We validated the differential phosphorylation of CREB by different means. Quantification of total and phosphorylated CREB by alphaLISA revealed an increase of the p-CREB/CREB ratio in *Ctsd*$^{-/-}$ cells during long-term starvation, while in *Ctsd*$^{+/+}$ cells it was rather mildly decreased (Supplementary Fig. 10a). As the amount of phosphorylated CREB in the nucleus is decisive for its function as a transcriptional activator, we went further and performed immunofluorescent stainings in 1% FCS and 1% FCS LT *Ctsd*$^{+/+}$ and *Ctsd*$^{-/-}$ cells (Supplementary Fig. 10b). Quantification confirmed the increase of the nuclear p-CREB/CREB ratio in

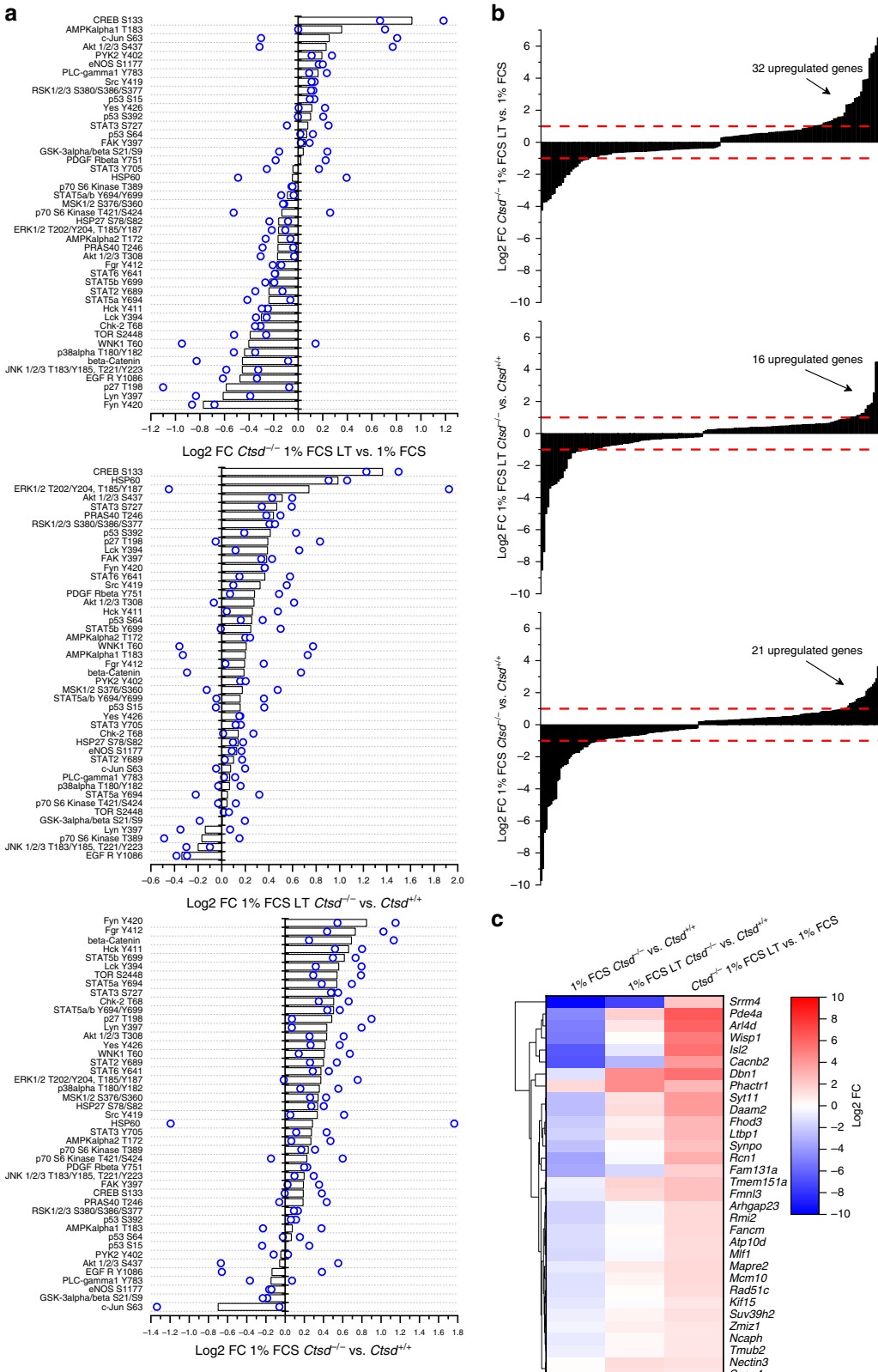

**Fig. 8 CTSD-deficient tumor cells having escaped quiescence activate tumorigenic CREB. a** Phospho-kinase antibody arrays with lysates from *Ctsd*[+/+] and *Ctsd*[−/−] PyMT cells cultured for 7–10 days (1% FCS) or for ≥8 weeks (1% FCS LT) in 1% FCS medium. Changes in phosphorylation are plotted as log2 fold change (FC) for indicated comparisons (*n* = 2 independent experiments). **b**, **c** Transcriptome analysis of 1% FCS and 1% FCS LT *Ctsd*[+/+] and *Ctsd*[−/−] PyMT cells (*n* = 3 independent experiments). **b** Change in expression of significantly regulated genes of the CREB gene set for indicated comparisons. Red dashed lines, borders for gene up- and downregulation (|log2 FC | > 1). **c** Clustering analysis of the log2 FC of CREB genes with significant upregulation in *Ctsd*[−/−] 1% FCS LT versus 1% FCS PyMT cells for indicated comparisons shown as heatmap. Source data are provided as a Source Data file.

$Ctsd^{-/-}$ cells, while this ratio remained unaffected in long-term FCS-starved $Ctsd^{+/+}$ cells (Supplementary Fig. 10c). This difference in p-CREB levels translated into differential sensitivity to the CREB inhibitor KG-501. The $EC_{20}$, $EC_{50}$ and $EC_{80}$ values for KG-501 were reduced by approximately 20% in 1% FCS LT $Ctsd^{-/-}$ compared to 1% FCS LT $Ctsd^{+/+}$ cells (Supplementary Fig. 10d).

Taken together, the data provide evidence for higher CREB phosphorylation and a CREB-associated oncogenic gene expression pattern in long-term starved CTSD-deficient PyMT cells. In consequence, $Ctsd^{-/-}$ cancer cells are able to re-install growth in 1% FCS LT conditions despite the continued impairment of mTORC1 signaling. It remains to be investigated whether additional molecular pathways help cancer cells to adapt to the deficiency of this major lysosomal aspartic protease.

## Discussion

The lethal phenotype of constitutive CTSD knockout mice precluded in vivo tumor studies, thus hindering addressing the long-suspected role of CTSD in breast cancer by means of this approach. Using conditional CTSD knockout PyMT mice, we show that abrogating CTSD expression in mammary epithelial cells, but not in myeloid cells, delayed tumor development substantially. The PyMT antigen activates Ras and PI3K-Akt signal transduction[37]. Those pathways are very frequently deregulated in human breast cancer[38]. In PyMT cells, CTSD deficiency stalled tumor cell proliferation under short-term starvation by preventing the recruitment of mTORC1 and its activation at the lysosomal membrane resulting in enhanced autophagic flux and cellular quiescence. CTSD-deficient tumor cells were able to adapt to long-term FCS starvation as indicated by resumed proliferation. We argue that this phenomenon is in analogy to the regrowth of the primary CTSD-deficient tumors after a lag-phase of about two months. We identified CREB as one of the molecular players that mediate the development of cancer cell-intrinsic resistance to the CTSD knockout.

The novelty of our study is the investigation of cell type-specific functions of CTSD in vivo in a relevant primary breast cancer model. We provide evidence for a tumor cell-specific effect of CTSD deficiency on tumor development in PyMT mice. This is important, as macrophages possess large amounts of CTSD and thus constitute a major part of stromal CTSD, which was claimed to have prognostic relevance[20]. Based on our results, we would rather agree with studies assigning CTSD in tumor cells prognostic significance[39]. A growth-inhibitory effect of CTSD deficiency in tumor cells has also been described by others[40]. In MDA-MB-231 breast cancer cells this is probably due to the lack of mitogenic pro-CTSD secretion, although those knock-down studies could not discriminate between intra- and extracellular CTSD functions. In contrast to that, our competitive growth assays suggest a secretion-independent mechanism. CTSD knockout PyMT cells showed a clear growth disadvantage in vitro and in vivo despite being surrounded by CTSD-expressing PyMT or stromal cells, respectively. From this, we infer an intrinsic proliferation block in tumor cells lacking CTSD.

Experiments with constitutive CTSD knockout tumor cells support this hypothesis. Under limited nutrient supply, CTSD deficiency induced high β-galactosidase activity, resulted in reduced proliferation, and increased sensitivity to the senolytic agent Navitoclax. Together with apoptosis, cellular quiescence, which is characterized by a reversible cell cycle-arrest, controls growth and maintains tissue homeostasis[41]. We think that cellular quiescence is the reason why CTSD-deficient tumors were blocked in their development for two months. Strikingly, macroautophagy, a stress response closely related to quiescence, was

triggered[41]. Starved CTSD-deficient cells showed an increased autophagic flux, the rate at which autophagy substrates are recognized, segregated and degraded. This resulted in high levels of free amino acids as evidenced by targeted amino acid analysis by LC–MS/MS of whole cell lysates. In this respect, it has been shown that autophagy, which is stimulated by mTORC1 inhibition, can positively regulate quiescence and thus suppress malignant transformation of normal cells[41]. Lysosomal and cytosolic amino acids usually activate Rag GTPases associated with the Ragulator complex on the lysosomal membrane resulting in mTORC1 recruitment to the lysosomal surface as prerequisite for mTORC1 kinase activity[33]. We provide evidence that major components recruiting mTORC1 to the lysosome as well as the proteins that activate lysosome-associated mTOR are present at normal levels in $Ctsd^{-/-}$ cells. We also show by mLST8-GFP pull-down experiments that mTORC1 is able to assemble properly in CTSD-deficient cells. To our surprise, mTOR co-localization with the lysosomal membrane protein LAMP1 was strongly reduced in CTSD-deficient cells, despite the presence of high levels of free amino acids. Also, experiments with amino acid starvation and re-addition of amino acids had very little effect on the disturbed association of mTOR with the lysosome. As mTORC1 tethering to the lysosome is a key determinant for its activity, we suspected impaired mTOR signaling in the CTSD knockout. Indeed, phosphorylation of a major mTORC1 target, namely the P70S6K, was reduced indicating impaired mTORC1 activity in CTSD-deficient tumor cells. As integrator of proliferation signals and the internal metabolic state of cells mTORC1 decides in favor of cell growth and division or metabolic adaptation, e.g., by induction of autophagy. Taken together, all our cellular and molecular readouts strongly suggest disturbed lysosomal mTORC1 localization and therefore a strongly reduced mTOR signaling. Up to now, we cannot explain how CTSD deficiency affects the assembly of the mTORC1 multiprotein complex at the lysosomal surface. As described above, there is no lack of amino acids or individual proteins required for mTORC1 assembly or activity. However, patients with inherited CTSD deficiency develop neuronal ceroid-lipofuscinosis type 10, which is characterized by an altered lipid metabolism in the postmitotic neurons[7,8]. We assume that also in $Ctsd^{-/-}$ cancer cells changes in lysosomal lipid composition hinder the fine-tuning of mTORC1 assembly at the lysosomal surface.

Continued exposure of CTSD-deficient tumor cells to nutrient restriction did not culminate in cell death, in contrast to neurons that already succumb to death by CTSD deficiency alone. Instead, proliferation and tumor development were resumed after a rather long lag-phase. While mTORC1 signaling remained impaired, lysosomal mass and autophagic flux were brought back to basic levels. The ability of the tumor cells to adapt to such a stress condition let us hypothesize that they acquired some sort of plasticity while being stalled in quiescence. Importantly, we could exclude the occurrence of mutations in oncogenes or tumor suppressors by whole-exome sequencing of $Ctsd^{-/-}$ tumors. Rather we found that CTSD-deficient tumor cells that escaped quiescence changed from an epithelial to a mesenchymal-like morphology, suggesting EMT. Indeed, CTSD-deficient cells, which had resumed proliferation, downregulated keratins, desmosome and tight junction components and upregulated EMT-inducing transcription factors as well as typical mesenchymal marker proteins.

Regarding the escape from quiescence, we show by RNA-Seq a compensatory transcriptional activation of oncogenic signaling such as the Ras-ERK and PI3K-Akt pathways. One downstream target of these kinases specifically activated by phosphorylation in CTSD-deficient previously quiescent tumor cells was CREB. Overexpression of this transcription factor has been linked to

tumorigenesis and resistance to Raf/MEK/ERK and PI3K/Akt inhibitors[42]. Interestingly, tumor cells with chronic PI3K/mTOR inhibition are able to resume proliferation through CREB stabilization[43]. Furthermore, progression of hypoxic tumors in PyMT mice is accompanied by CREB activation[44]. Based on its activating phosphorylation and transcriptional induction of its target genes, we think that CREB is important for CTSD-deficient tumor cells to overcome the proliferation block during starvation. In a recent paper Jewell et al.[45] report mTORC1 inhibition downstream of GPCR signaling through PKA-mediated Raptor phosphorylation. Concomitantly, PKA also phosphorylated CREB. However, in this report Raptor phosphorylation did not block mTORC1 association with the lysosome upon amino acid stimulation as we observed in $Ctsd^{-/-}$ PyMT cells. In addition, our data show mTORC1 inactivation preceding the increase of CREB phosphorylation. Therefore, mTORC1 in CTSD-deficient PyMT cells is unlikely to be actively downregulated by PKA-mediated Raptor phosphorylation, while we cannot exclude a contribution of PKA signaling to CREB activation in the long term.

In summary, our data demonstrate that deficiency for a major lysosomal proteinase, i.e., CTSD, interferes with a dominant oncogenic signaling pathway in breast cancer, i.e., PI3K-mTOR signaling. Initially, attenuated mTORC1 output results in reduced proliferation and a quiescent phenotype of the cancer cells. However, in contrast to the CTSD-deficient postmitotic neurons affected in CLN10, breast cancer cells appear to be able to rewire proliferative and survival signaling, e.g., by CREB activation, thereby resuming malignant growth and metastasis. Nevertheless, one could therapeutically exploit the CTSD deficiency-induced phenotype by applying a two-hit synthetic lethal strategy as suggested by Wang et al.[46], with initial induction of a quiescent cancer cell phenotype by CTSD inactivation, followed by specific killing with senolytic agents.

## Methods

**Mice**. All animal procedures were approved by the legal authorities and ethics committee at the regional council Freiburg (registration number: G14/18) and were performed in accordance with the German law for animal welfare. Animals were kept at 21–23 °C, 45–60% humidity and a 12 h dark/12 h light cycle. All mouse strains, except for immunocompromised mice, were maintained on a C57BL/6 N background. Mice heterozygous or homozygous for the untargeted $Ctsd$ allele (+) or the conditional $Ctsd$ allele flanked by loxP sites (fl) (MGI:5702318) are CTSD-competent and summarized as Controls. For myeloid- and mammary epithelium-specific deletion of CTSD ($Ctsd^{-/-}$) the $LysM-cre$ deleter strain from Jackson Laboratory (B6.129P2-Lyz2$^{tm1(cre)Ifo}$/J) or the $MMTV-cre$ deleter strain from Prof. Dr. Rolf Kemler (Max Planck Institute of Immunobiology and Epigenetics, Freiburg, Germany)[47] were used, respectively. We visualized in vivo cre-mediated recombination using a global double-fluorescent cre reporter mouse (C57BL/6J-Tg (pCA-mT/mG)), abbreviated as $mTmG$[24]. As tumor model we used $MMTV-PyMT$ mice expressing the polyoma virus middle T oncogene (PyMT) under the control of the mouse mammary tumor virus ($MMTV$) LTR promoter[22], abbreviated as PyMT mice. Tumor growth was monitored by palpation, twice per week from 8 weeks onwards until tumor harvest at indicated time points or at end-stage size (1 cm³). Female BALB/c-Rag2$^{-/-}$;γc$^{-/-}$ mice deficient for lymphocytes and NK cells[48,49] were used for orthotopic transplantations.

**Ultrasound imaging**. Eighteen-week-old tumor-bearing mice were anesthetized under spontaneous breathing conditions with isoflurane (1.5%, 0.8 l/min) and immobilized on a dedicated animal platform with a built-in animal monitoring system. The respiration rate was continuously monitored and kept at a constant level. The two largest tumors were depilated to reduce imaging artefacts and respiratory gating was used to reduce motion artefacts during the scan. Images were acquired with a Vevo 3100 preclinical ultrasound imaging system (FUJIFILM VisualSonics) using B-mode and a 3D volume scan. Tumor volumes were calculated using the Vevo LAB software (V3.1.1).

**Tissue preparation and histology**. For whole mounts, the left inguinal mammary gland was prepared from 8-week-old PyMT mice. Tissues were fixed in 4% paraformaldehyde (PFA) for 2 h at 4 °C and stained in carmine alum solution (0.2% carmine and 0.5% aluminum potassium sulfate in $H_2O$) overnight at room

temperature (RT). Subsequently, tissues were dehydrated in ethanol (70%, 95% and 100% for 1 h each), cleared in xylene overnight and stored in methyl salicylate. For immunohistochemistry tissues were fixed in 4% PFA, paraffin-embedded, sectioned at 5 μm, and stained for Ki67 (Abcam, ab15580; 1:400) or estrogen receptor alpha (Santa Cruz Biotechnology, sc-8005; 1:30) as well as with Mayer's hemalum solution (Merck, 109249). Detection of primary antibody was performed using the Vectastain Elite ABC kit (Vector Laboratories), followed by diaminobenzidine (Sigma, D3939) incubation. For immunofluorescence tissues were fixed in 4% PFA, Tissue-Tek® O.C.T.™-embedded, sectioned at 7 μm, and stained with DAPI (Thermo Fisher, D1306). Images were captured continuously with a 4× objective and merged into a single macro image or captured with a 40× objective using a Keyence BZ-9000 device and software. Adjustment of brightness and contrast, as well as measurement of number and area of metastatic foci and dysplastic lesions, were performed using ImageJ (Fiji, National Institutes of Health) software.

**Cell culture, transfection, transduction and treatments**. Bone marrow cells from tibia and femur were collected and pelleted (290× $g$ for 10–15 min). After erythrocyte lysis (155 mM NH₄Cl, 10 mM KHCO₃, and 0.1 mM EDTA), cells were washed with PBS and cultured in macrophage differentiation medium (RPMI 1640 + GlutaMAX supplemented with 15% GM-CSF-enriched cell-conditioned medium, 10% FCS, 5% horse serum, 5% Penicillin–Streptomycin, 1% sodium pyruvate, and 0.04% β-Mercaptoethanol (1% in $H_2O$)) in Teflon® bags at 37 °C with 5% CO₂. After 10 days macrophages were loosened by kneading the bags, resuspended in DMEM/F12 medium supplemented with 10% FCS, 2 mM L-glutamine, and 1% Penicillin–Streptomycin, and cultured on plastic.

Tumor cell lines were generated from end-stage tumors of $Ctsd^{fl/fl};mTmG$, $Ctsd^{+/+}$ and $MMTV-cre;Ctsd^{-/-}$ ($Ctsd^{-/-}$) PyMT mice. To this end, tumors were mechanically disrupted and enzymatically digested in DNase I (0.25 mg/ml), Hyaluronidase I-S (1 mg/ml) and Collagenase IV (6 mg/ml) (5 ml each). The resulting suspension was filtered twice using 100 and 40 μm cell strainers and washed twice with PBS. These freshly isolated cells were then cultured until spontaneous immortalization before performing experiments. PyMT cells were cultured in DMEM/F12 medium supplemented with 10% FCS, 2 mM L-glutamine, and 1% Penicillin–Streptomycin at 37 °C with 5% CO₂, a condition referred to as 10% FCS. For FCS starvation FCS levels were reduced to 1%, a condition that was applied to cells for <2 weeks (1% FCS) or for long-term, meaning ≥8 weeks (1% FCS LT). For acute amino acid stimulation, cells were starved in a special starvation medium (DMEM Low Glucose, w/o Amino Acids, Pyruvic Acid (USBiologicals, D9800-13) and 1% Penicillin–Streptomycin) for 30 min, followed by 30 min amino acid stimulation (special starvation medium supplemented with MEM Amino Acids Solution (PAN Biotech, P08-30100) and 2 mM L-glutamine).

$Ctsd^{fl/fl};mTmG$ tumor cells were stably transduced with a two-component Dox-inducible cre expression system[50] by two consecutive infections. For retrovirus production Plat-E cells were transfected with 8 μg DNA using polyethylenimine (PEI). Two days after transfection, Plat-E supernatant was filtered (0.45 μm), concentrated in case of pSH461MK-Cre with Lenti-X™ Concentrator, and complemented with polybrene to a final concentration of 8 μg/ml. Target cells were seeded 24 h before infection with the pSH582 construct, encoding the Dox-regulated transcriptional activator and suppressor proteins, followed by Blasticidin S (10 μg/ml) selection 24 h after infection. Drug-resistant cells were then infected with pSH461MK-Cre and selected with Puromycin (2 μg/ml). $Ctsd^{fl/fl};mTmG$ tumor cells transduced with pSH582 and pSH461MK-Cre are referred to as $rtTA-tTS;tetO-cre;Ctsd^{fl/fl};mTmG$ cells.

$Ctsd^{+/+}$ and $Ctsd^{-/-}$ PyMT cells were stably transduced with an EGFP or EGFP-mLST8 expression vector[51]. For retrovirus production HEK293 cells were transfected with pLVX-puro-EGFP or pLVX-puro-EGFP-mLST8, together with psPAX2 (packaging) and pMD2.G (envelope) plasmids (ratio 5:3:1.5, respectively) using Superfect. Two days after transfection, HEK293 supernatant was filtered (0.45 μm) and complemented with polybrene to a final concentration of 5 μg/ml. Cells were seeded 24 h before infection and selection with Puromycin (2 μg/ml) started 48 h after infection.

To induce cre expression cells were treated with 2 μg/ml Dox for 4 days or, in case of competitive growth assays, for 24 h. The senolytic agent Navitoclax (AdooQ BioScience, A10022) was incubated at 100 nM for 24 h with cells starved for 10 days, followed by analyzing apoptosis by flow cytometry. To inhibit mTOR signaling and induce lysosomal biogenesis, cells were treated with 500 nM Torin 1 (Tocris, 4247) for 1 h and overnight, respectively. For autophagic flux experiments, cells were treated with 3.3 nM Bafilomycin A1 (InvivoGen, tlrl-baf1) or a combination of 10 μM Pepstatin A (Sigma–Aldrich, P5318) and 10 μM E64d (Bachem, N-1650) additionally to starvation in DMEM Low Glucose, w/o Amino Acids, Pyruvic Acid (USBiologicals, D9800-13) for 0.5 or 6 h, respectively. In immunofluorescent staining experiments, Pepstatin A treatment started 3 days prior to starvation/ stimulation/ fixation and was refreshed after 48 h. The CREB inhibitor KG-501 (Selleckchem, S8409) was incubated at concentrations ranging from 0–50 μM for 24 h with 1% FCS LT $Ctsd^{+/+}$ and $Ctsd^{-/-}$ PyMT cells, followed by analyzing cell metabolic activity by MTT assay. Cell morphology was regularly checked and documented with pictures using a Nikon Eclipse TS100 microscope.

**Orthotopic transplantation**. Dox-pulsed $rtTA-tTS;tetO-cre;Ctsd^{fl/fl};mTmG$ cells (0.5 × 10⁶ cells in 50 μl of a 1:1 PBS-Cultrex™ (Trevigen, 343200501) mixture)

were transplanted into the right inguinal mammary gland of female Rag2$^{-/-}$;γc$^{-/-}$ mice via a 5 mm lateral incision. Tumor growth was monitored by palpation once a week until tumor harvest at end-stage size (1 cm$^3$). Tumors were processed as described for the generation of tumor cell lines, followed by flow cytometry analysis of primary tumor cells.

**Cell proliferation dye labeling**. Cells were labeled with the cell proliferation dye eFluor™ 670 (Thermo Fisher, 650840) following the manufacturer's recommendations and analyzed 72 h afterwards by flow cytometry. Unstained and freshly stained cells were used as controls.

**MTT assay**. Cells were seeded 24 h before treatment with the inhibitor or solvent control. After 24 h of treatment, cells were incubated 4 h with MTT-containing medium (0.5 mg/ml). Resulting formazan crystals were dissolved in DMSO and the absorbance was measured at 570 nm using an EnSpire multimode plate reader.

**Staining acidic compartments in live cells**. Cells were trypsinized and incubated with 0.1 μM LysoTracker™ Green (Invitrogen, L7526) and 1 μg/ml propidium iodide (PI) in FACS buffer (PBS, 2% FCS, and 5 mM EDTA) for 15 min at 37 °C. After washing twice with PBS cells were resuspended in FACS buffer and analyzed by flow cytometry.

**Flow cytometry**. Cells were resuspended in FACS buffer (PBS, 2% FCS, and 5 mM EDTA) or Annexin-V-binding buffer (0.1 M HEPES (pH 7.4), 1.4 M NaCl, and 25 mM CaCl$_2$). For apoptosis assay cells at a concentration of $1 \times 10^6$ cells/ml were incubated with Annexin-V-FITC (BD, 556419) for 15 min at RT and subsequently with 7-AAD (eBioscience, 00699350) for 5 min at RT at appropriate dilutions. Analysis was performed using a LSR II flow cytometer with FACSDiva and FlowJo software (all BD Bioscience). Cell debris and doublets were excluded using forward and side scatter. Further analysis depended on the experiment (see Supplementary Fig. 11 for all gating strategies). For competitive growth assays the ratio of Tomato$^+$ (non-recombined) cells to Tomato$^+$GFP$^+$ and GFP$^+$ (recombined) cells was assessed. For apoptosis assays the amount of Annexin-V$^-$7-AAD$^-$ (living) cells relative to DMSO-treated control and normalized to Ctsd$^{+/+}$ cells is given; staurosporine-treated cells served as positive control for Annexin-V binding and 7-AAD uptake. For label-retention assays eFluor™ 670 intensity in PI$^-$ cells was divided into negative, low, and high to distinguish between fast-, slow-, and non-proliferating (label-retaining) cells, respectively. For the assessment of acidic compartments in live cells, LysoTracker™ Green intensity in PI$^-$ cells was divided into negative, low, medium, and high. For lysosomal activity assays the recovery of fluorescence from the self-quenched substrate was quantified by the median fluorescence intensity.

**Acidic β-galactosidase staining**. Staining for acidic β-galactosidase was performed in cells grown on collagen-coated coverslips (Neuvitro Corporation, GG-12-15-Collagen) according to the manufacturer's recommendations (Cell Signaling, 9860). Cells were imaged with a 40× objective using the Keyence BZ-9000 device.

**Immunofluorescence**. 10% FCS, 1% FCS, FCS- and amino acid-starved, and acutely amino acid-stimulated Ctsd$^{+/+}$ and Ctsd$^{-/-}$ cells were seeded on collagen-coated coverslips (Neuvitro Corporation, GG-12-15-Collagen). After fixation in a 1:1 mixture of 4% PFA in PBS and culture medium for 5 min, followed by fixation in 4% PFA in PBS for 15 min at 4 °C, cells for CREB and p-CREB staining were additionally treated with 100% methanol for 10 min at −20 °C. Permeabilization for all cells was done with 0.2% Saponin in PBS for 5 min with shaking, unspecific antibody binding was blocked with 0.1% Saponin/1% BSA in PBS for 20 min. Primary antibodies for p-CREB (Cell Signaling, 9198; 1:200), CREB (Cell Signaling, 9197; 1:200), LAMTOR4 (Cell Signaling, 12284; 1:1000), LC3 (Cell Signaling, 2775; 1:200), LAMP1 (Abcam, ab25245; 1:750), and mTOR (Cell Signaling, 2983; 1:200) were applied for 1.5 h at RT or overnight at 4 °C. Subsequently, coverslips were washed and incubated with the corresponding secondary antibodies donkey-anti-rabbit-AF488 (Life Technologies, A21206; 1:1000) and goat-anti-rat-AF594 (Life Technologies, A11007; 1:1000) for 45 min at RT. After staining with Hoechst (Sigma, B2261) for 3 min, coverslips were mounted with PermaFlour. Double immunostainings were imaged in a confocal fluorescence microscope (Leica SP8, 63× objective) using Leica LAS software. Images were processed, analyzed, and exported using the Volocity software package (PerkinElmer). The Pearson correlation coefficient for mTOR/LAMP1 co-localization was determined with the Volocity co-localization tool after setting uniform thresholds across all conditions. The average co-localization was quantified across at least 5 images from 2 independent experiments. CREB and p-CREB immunostainings were imaged in an Axio Observer.Z1 fluorescence microscope (Zeiss) and evaluated with ImageJ (Fiji, National Institutes of Health). The mean gray value of CREB and p-CREB immunofluorescence was determined in each nucleus per image. The median of the mean gray values was quantified for each condition in 3 independent experiments. Subsequently, the ratio of p-CREB to CREB was calculated.

**RNA isolation and RT-PCR**. Total RNA from snap-frozen tumor tissue or tumor cells was isolated and purified using the RNeasy Mini kit (Qiagen). cDNA was generated using the iScript™ cDNA synthesis kit (BioRad) and analyzed by real-time PCR (RT-PCR) in a CFX96 Real-Time system (BioRad) using SYBR Select master mix for CFX (Applied Biosystems) with the following intron-spanning primer pairs: beta actin (Actb), forward 5′-ACCCAGGCATTGCTGACAGG-3′, reverse 5′-GGACAGTGAGGCCAGGATGG-3′; cathepsin L (Ctsl), forward 5′-GCACGGCTTTTCCATGGA-3′, reverse 5′-CCACCTGCCTGAATTCCTCA-3′; lysosomal-associated membrane protein 1 (Lamp1), forward 5′-GTGACAGG TTTGGGTCTGTGGA-3′, reverse 5′-GGTCTGATAGCCGGCGTGAC-3′; microtubule-associated protein 1 light chain 3 alpha (Map1lc3a = Lc3a), forward 5′-AGCTTCGCCGACCGCTGTAA-3′, reverse 5′-CGGCGCCGGATGATCTT GAC-3′; N-Cadherin (Cdh2), forward 5′-TATATGCCCAAGACAAAGAAACC-3′, reverse 5′-TTGGCAAGTTGTCTAGGGAATAC-3′; p16 (Cdkn2a/p16), forward 5′-GGCGCTTCTCACCTCGCTTG-3′, reverse 5′-GCCCATCATCATCACCTGG TCC-3′; p21 (Cdkn1a/p21), forward 5′-CCAGCAGAATAAAAGGTCCACA-3′, reverse 5′-ACCGAAGAGACAACGGCACA-3′; MPyVgp2 middle T antigen (PyMT), forward 5′-TCCAACAGATACACCCGCAC-3′, reverse 5′-GGTCT TGGTCGCTTTCTGGA-3′; sequestosome 1 (Sqstm1 = p62), forward 5′-GTCAG-CAAACCTGACGGGGC-3′, reverse 5′-CCGGGGATCAGCCTCTGTAGAT-3′; and Vimentin (Vim), forward 5′-TCCCTTGTTGCAGTTTTTCC-3′, reverse 5′-GATGAGGAATAGAGGCTGCC-3′. The average of technical replicates was normalized to Actb using the delta Ct method.

**Protease activity assay**. Cells were resuspended in sodium acetate buffer (100 mM sodium acetate, 1 mM EDTA, 0.05% Brij 35, and 1 mM DTT; pH 4.0 or pH 5.5 for aspartic or cysteine protease activity, respectively) and mechanically disrupted using a tight dounce homogenizer. 25 μM of the fluorogenic substrate Mca-Gly-Lys-Pro-Ile-Leu-Phe-Phe-Arg-Leu-Lys(Dnp)-D-Arg-NH$_2$ (Bachem, M-2455) or Z-Phe-Arg-AMC (Bachem, I-1160) was incubated at 40 °C or 37 °C with cell lysates to assess aspartic or cysteine protease activity, respectively. Using an EnSpire multimode plate reader emission of fluorescence was measured every minute for 1 h and 45 min at 393 nm or 460 nm (excitation at 328 nm or 360 nm) for aspartic or cysteine protease substrates, respectively. Enzyme activity was normalized to protein concentration of the samples.

**Lysosomal activity assay**. Lysosomal activity was determined with the Lysosomal Intracellular Activity Assay Kit (Abcam, ab234622) following the manufacturer's recommendations. Cells incubated with 1× Cytochalasin D (included in Assay Kit), 50 nM Bafilomycin A1 (InvivoGen, tlrl-baf1) or without any substrate served as controls. Cells were analyzed with a BD™ LSR II flow cytometer.

**Western blot**. Lysates of cells or tumor tissue containing 20–40 μg protein were subjected to SDS-PAGE and transferred to a PVDF membrane (GE Healthcare) via wet blot system (BioRad). Membranes were blocked with 3% BSA in TBS-Tween (0.1%) and incubated with primary antibodies for ACTB (MP, 691001; 1:2000), CDH2 (Cell Signaling, 4061; 1:500), CTSB (R&D, BAF965; 1:200), CTSD (R&D, AF1029; 1:200), CTSL (R&D, AF1515; 1:200), GFP (Cell Signaling, 2555; 1:1000), LAMP1 (Cell Signaling, 3243; 1:500), LAMTOR4 (Cell Signaling, 12284; 1:1000), LC3 (Cell Signaling, 2775; 1:500), mTOR (Cell Signaling, 2983; 1:1000), p-p38 MAPK (Cell Signaling, 9211; 1:1000), p38 MAPK (Cell Signaling, 9212; 1:1000), p62 (Cell Signaling, 5114; 1:500), p-P70S6K (Cell Signaling, 9205; 1:500), P70S6K (Enzo, ADI-KAP-CC035-E; 1:250), RHEB (Cell Signaling, 13879; 1:1000), RPTOR (Cell Signaling, 2280; 1:1000), RRAGC (Cell Signaling, 5466; 1:1000), p-S6 (Cell Signaling, 4857; 1:1000), S6 (Cell Signaling, 2317; 1:1000), TUBA (Sigma, T9026; 1:1000), and VIM (BD, 550513; 1:500) mostly overnight at RT. Membranes were washed, incubated with the corresponding secondary antibodies goat-anti-mouse-peroxidase (Sigma, A0168; 1:5000), goat-anti-rabbit-horseradish peroxidase (BioRad 172-1019; 1:5000), and rabbit-anti-goat-peroxidase (Sigma, A5420; 1:5000) for 45–60 min at RT, washed again, and developed using West Pico/Femto Chemiluminescent Substrate (Thermo Scientific) or PURECL Dura Substrate (Vilber Lourmat, PU4400500). Signals were detected and analyzed using a Fusion SL Detection System and FusionCapt Advance software (Vilber Lourmat). Whenever signals from phosphorylated and total proteins were detected on separate membranes, they were first normalized to the corresponding loading control before building ratios of phosphorylated to total protein. Ratios of LC3-II/LC3-I and p-P70S6K/P70S6K are given relative to 10% FCS Ctsd$^{+/+}$.

**Co-immunoprecipitation**. Cells were lysed in CHAPS buffer (40 mM HEPES (pH 7.5), 120 mM NaCl, 1 mM EDTA, 10 mM sodium pyrophosphate, 10 mM ß-glycerophosphate, 1.5 mM Na$_3$VO$_4$, 0.3% CHAPS, one tablet phosphatase inhibitors (Roche) and one tablet EDTA-free protease inhibitors (Roche)). EGFP-fusion proteins were immunoprecipitated from cell lysates using GFP-Trap beads (Chromotek, gta-20) following the manufacturer's instructions. The immunoprecipitates were fractionated by SDS-PAGE and co-immunoprecipitated proteins were detected by Western blot.

**Phospho-kinase antibody array**. Lysates from 1% FCS and 1% FCS LT $Ctsd^{+/+}$ and $Ctsd^{-/-}$ cells were prepared and incubated with membranes spotted with antibodies specific for kinase phosphorylation sites (R&D, ARY003B) following the manufacturer's instructions. Signals were detected using a Fusion SL Detection System (Vilber Lourmat) and analyzed with HLImage++ software (R&D).

**AlphaLISA and AlphaLISA SureFire Ultra assays**. CREB, phosphorylated CREB and IL-6 levels in cells and cell-conditioned media were measured with the AlphaLISA® SureFire® Ultra™ CREB Total Assay Kit (ALSU-TCREB-A500), AlphaLISA® SureFire® Ultra™ p-CREB (Ser133) Assay Kit (ALSU-PCREB-A500), and the AlphaLISA Mouse Interleukin 6 (mIL6) Kit (AL504 C) from PerkinElmer, respectively. The assays were performed following the manufacturer's recommendations in a 384-well OptiPlate (PerkinElmer, 6007290). Cells were seeded in 24-well plates and cultivated for 48 h in 500 μl medium. Cell-conditioned media were harvested at equal cell confluency and IL-6 AlphaLISA was performed. For p-CREB and CREB AlphaLISA, cells were harvested in 200 μl 1× AlphaLISA SureFire Ultra lysis buffer. AlphaLISA SureFire Ultra assay results were normalized to the protein concentration of the respective cell lysate as determined by Pierce™ BCA™ Protein-Assay (Thermo Fisher, 23225).

**Targeted amino acid analysis by LC–MS/MS**. 1% FCS $Ctsd^{+/+}$ and $Ctsd^{-/-}$ cells were washed three times with 0.9% NaCl before being detached on ice by scraping in methanol:water (90:10) containing 1 μg/ml O-Methyl-L-tyrosine (Alfa aesar, H63096.03) as internal standard. Cells were lysed using glass beads and a Precellys tissue homogenizer ($3 \times 15$ s at 6500 rpm, 10 s break, $-10$ °C). Homogenized lysates were centrifuged ($21,000\times g$, 10 min, 4 °C) and 1 ml of metabolite-containing supernatant was prepared for LC–MS/MS analysis (dried, resuspended in 100 μl water and centrifuged ($20,000\times g$, 5 min, 4 °C)), whereas the pellet was resuspended in the remaining supernatant for fluorometric DNA quantification (20 μl were incubated with 80 μl Hoechst solution (20 μg/ml) for 30 min at RT and emission of fluorescence was measured at 460 nm (excitation at 346 nm) using an EnSpire multimode plate reader[52]. 80 μl of each metabolite-containing supernatant were transferred to LC–MS glass vials containing inserts for small volume injection (Agilent Technologies). Injection volume was set to 5 μl. 20 μl of each supernatant were used to prepare a pool sample serving as quality control. Samples were injected in randomized order. LC–MS/MS analysis of the extracted metabolites was performed on a 1290 Infinity UHPLC system coupled to a 6460 triple quadrupole mass spectrometer (Agilent Technologies) via an electrospray ionization source (ESI Jetstream, Agilent Technologies). For separation of amino acids a Waters Acquity UPLC BEH Amide column (150 mm × 2.1 mm, 1.8 μm) was used at 50 °C and a flow rate of 0.6 ml/min with water $+0.1\%$ formic acid as buffer A and acetonitrile (Roth, AE70.2) $+0.1\%$ formic acid as buffer B. The following gradient was applied: 0.0–0.1 min at 90% B, 0.1–0.2 min to 85% B, 0.2–1.0 min to 75% B, 1.0–2.0 min to 40% B. Finally, the column was washed for 3 min at 50% B and re-equilibrated for 4 min at 90% B. The total run time was set to 9 min. The following MS settings were applied: capillary voltage, 4000 V; nozzle voltage, 500 V; gas temperature, 300 °C; gas flow, 7 l/min; sheath gas temperature, 350 °C; sheath gas flow, 11 l/min; nebulizer pressure, 50 psi. MS/MS spectra were acquired in dMRM mode. Mass spectral transitions were optimized using the built-in MassHunter Optimizer tool (Agilent Technologies, Waldbronn, Germany)[53]. LC–MS/MS data analysis was performed using MassHunter Quant B07 (Agilent Technologies, Waldbronn, Germany) and results were exported to CSV table for further processing[53]. Following data export peak areas were normalized to the internal standard Methyl-tyrosine and to DNA content of the sample, followed by blank (sample without cells) subtraction. Using MetaboAnalyst[54] each variable was mean-centered and divided by the value range of each variable, called range scaling. The resulting z-scores were used for PCA and hierarchical cluster analysis.

**Whole-exome sequencing**. Genomic DNA was isolated from tumor pieces (≤10 mg) and tail lysates (germline control) of 18-week-old control and *MMTV-cre; Ctsd^{-/-}* PyMT mice (4 each) using the QIAamp DNA Micro kit (Qiagen). DNA integrity was verified using a TapeStation system. The Agilent SureSelect XT Mouse AllExon Capture Kit was used for library preparation and afterwards sequenced on the Illumina HiSeq4000 platform with a read length of 100 base pairs (bp). The data were processed with Trimmomatic[55] and mapped to the GRCm38/mm10 genome assembly with the burrows-wheeler aligner (bwa-mem)[56]. For somatic variant calling VarScan2[57] was used. The identified variants include single nucleotide variants (SNV), insertions or deletions (Indel) up to 30 bp length, and loss of heterozygosity (LOH), and are further annotated with ANNOVAR[58]. Genes highlighted in circos plots represent murine homologs of frequently mutated genes in human breast cancer, defined as the top 10 mutated genes listed in the COSMIC database[59] for breast carcinoma (3/2018).

**RNA sequencing**. RNA was isolated from snap-frozen tumors of 18-week-old control and *MMTV-cre;Ctsd^{-/-}* PyMT mice ($n = 4$), and from 1% FCS and 1% FCS LT $Ctsd^{+/+}$ and $Ctsd^{-/-}$ tumor cells ($n = 3$) using the Universal RNA purification kit (Roboklon). RNA integrity was verified using a Bioanalyzer. RNA libraries were prepared from 25 ng tissue RNA and 1400 ng cell RNA and sequenced with 50 Mio reads using the Illumina HiSeq4000 platform. Raw FastQ

data were trimmed with Trimmomatic and mapped to the mouse GRCm38/mm10 genome assembly using STAR[60]. Differentially expressed genes were determined using the R/Bioconductor[61,62] package limma[63] for the following comparisons: *MMTV-cre;Ctsd^{-/-}* versus control ($p$-value of 0.1 without multiple testing correction), 1% FCS $Ctsd^{-/-}$ versus 1% FCS $Ctsd^{+/+}$ (1% FCS $Ctsd^{-/-}$ vs $Ctsd^{+/+}$), 1% FCS LT $Ctsd^{-/-}$ versus 1% FCS LT $Ctsd^{+/+}$ (1% FCS LT $Ctsd^{-/-}$ vs $Ctsd^{+/+}$), and 1% FCS LT $Ctsd^{-/-}$ versus 1% FCS $Ctsd^{-/-}$ ($Ctsd^{-/-}$ 1% FCS LT vs 1% FCS) ($p$-value of 0.05 with multiple testing correction). The latter comparison was corrected for expression changes in $Ctsd^{+/+}$ cells due to long-term cultivation in 1% FCS medium by following arithmetic operation: (1% FCS LT $Ctsd^{-/-}$—1% FCS LT $Ctsd^{+/+}$) versus (1% FCS $Ctsd^{-/-}$—1% FCS $Ctsd^{+/+}$). Analysis of the CREB gene set, which was downloaded from the Harmonizome database[36], was performed with adj. $p$-value < 0.1. Using STRING[35] a protein association network analysis with MCL clustering and a GSEA were performed on significantly deregulated genes (adj. $p$-value < 0.05, |log2 FC | > 1, |expression | > 1.3) for 1% FCS LT $Ctsd^{-/-}$ versus 1% FCS $Ctsd^{-/-}$. The 'lysosomal biogenesis and function' gene set was taken from Sardiello et al.[64] and modified by adding '$Ctsl$'. Log2 FC are plotted for all genes of this gene set, independent of $p$-values. A GSEA for autophagy-related GO pathways was performed using gage. All pathways containing 5–500 genes are plotted with their corresponding $q$-value.

**Data presentation and statistics**. Data in line and bar charts were expressed as the mean + SD, if not stated differently. $n$ represents independent experiments (biological replicates). Statistical analyses were carried out with OriginPro 2018G-2020 (OriginLab). Significance in the Kaplan–Meier analyses was calculated using the two-sided log-rank test. Significance of lesion area, in vitro and in vivo cell growth, cell viability, mRNA levels, protein levels, and Pearson correlation coefficients between CTSD-competent and CTSD-deficient settings was determined by two-sided one-sample/two-sample $t$-test. mRNA levels normalized to $Actb$ and protein levels normalized to 10% FCS $Ctsd^{+/+}$ were log-transformed before statistical analysis. Statistical significance of tumor volumes was calculated by the non-parametric Mann–Whitney test (two-sided). A two-sided three-way ANOVA was performed for the factors intensity of LysoTracker™, $Ctsd$ genotype and FCS. PCA was used as unsupervised multivariate analysis to visualize global alterations between $Ctsd^{+/+}$ and $Ctsd^{-/-}$ datasets. Hierarchical cluster analyses were performed using group average clustering method with Euclidean distance and plotted as heatmap with dendrogram.

**Reporting summary**. Further information on research design is available in the Nature Research Reporting Summary linked to this article.

## Data availability

The exome and transcriptome data have been deposited in the BioProject database under the accession code PRJNA663291 and in the Gene Expression Omnibus (GEO) database under the accession code GSE133328, respectively. The CREB gene set and the list of mutated genes in human breast carcinoma referenced during the study are available in public repositories from the harmonizome (http://amp.pharm.mssm.edu/Harmonizome/gene_set/CREB/MotifMap+Predicted+Transcription+Factor+Targets) and COSMIC (https://cancer.sanger.ac.uk/cosmic/browse/tissue?hn=carcinoma&in=t&sn=breast&ss=all) website, respectively. The mice used in this study and tissue or cells derived thereof are available upon request after completion of an appropriate material transfer agreement. All the other data supporting the findings of this study are available within the article and its supplementary information files and from the corresponding author upon reasonable request. A reporting summary for this article is available as a Supplementary Information file. Source data are provided with this paper.

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

## Acknowledgements

The authors thank Nicole Klemm, Susanne Dollwet-Mack and Ulrike Reif for their continuous expert technical assistance. Further, we would like to thank the Life Imaging Center (LIC) of the University of Freiburg for their support in cell imaging and the team of the Genomics and Proteomics Core Facility, German Cancer Research Center (DKFZ), Heidelberg, Germany, for their sequencing service. This work was financially supported by the Deutsche Forschungsgemeinschaft (DFG) SFB 850 subprojects B4 (T.B.), B7 (T.R. and C.P.), C9 and Z1 (M.B.), as well as Z2 (W.R.). The work was further supported by the

DFG grant (RE1584/6-2 to T.R.), a DFG-Heisenberg professorship (to T.B.) and a DFG Emmy Noether grant (STE2310/1-1 to F.S.). The German Cancer Consortium (DKTK) program Oncogenic Pathways supported with project L625 (to M.B.) and L627 (to T.R., C.P. and T.B.). Additional support was received from the German Federal Ministry of Education and Research (BMBF) within the framework of the e:Med research and funding concept CoNfirm (FKZ 01ZX1708F to M.B.) and by MIRACUM within the Medical Informatics Funding Scheme (FKZ 01ZZ1801B to P.M. and M.B.).

## Author contributions

S.K. and T.R. designed the study and wrote the manuscript. S.K., J.M., A.K., N.B. and T.R. performed experiments and/or analyzed data. M.H., P.M. and M.B. performed whole-exome sequencing-, RNA sequencing- and bioinformatics analysis. M.S. and B.K. performed the targeted amino acid analysis. W.R. provided expertise for the ultrasound imaging of primary breast tumors. F.S. performed the quantitative immunofluorescence co-localization studies and provided EGFP-mLST8 expression constructs. T.B. provided inducible expression systems. T.R., T.B. and C.P. created the scientific strategy for the project. All authors read, critically revised, and approved the final version of the manuscript.

## Funding

## Competing interests

The authors declare no competing interests.
