## [Peer Review File · Nature Communications]

Reviewers' comments:

Reviewer #1 (Remarks to the Author):

The manuscript by Ketterer et al entitled 'Cathepsin D deficiency in mammary epithelium transiently stalls breast cancer by interference with mTORC1 signaling' is a very well written and well-presented manuscript on a very timely and interesting topic. The authors use complementary in-vitro and in-vivo techniques to describe how cathepsin D (CTSD) expression represents a competitive advantage to breast cancer cells and significantly accelerates tumour development in the PyMT model. The authors explore the underlying mechanisms via which CTSD impacts breast epithelial cell proliferation using inducible KO of cathepsin D. They show that culturing the cells in low serum conditions leads to exit from the cell cycle (and induction of senescence) mediated by reduced mTORC1 activity, and a concomitant elevation of autophagy and expansion of the lysosomal compartment. Long term culture of *ctsd*^{-/-} cells in vitro (and following a latent period in vivo), they re-enter the cell cycle, mediated not via accumulation of genetic mutations but rather activation of growth pathways such as PI3K/Akt and ERK signalling. In particular, the authors identify CREB phosphorylation as an important event. Thus, they demonstrate tumour-cell specific role of CTSD and provide new insights into the role of CTSD in breast cancer.

Overall, I think the study is extremely interesting and most notably it clarifies an important issue in the field regarding the role of CTSD in breast cancer cells (as opposed to myeloid cells).

I do however have some fundamental questions regarding the mechanisms that the authors propose. If the *ctsd*^{-/-} cells proliferate, even slowly, in low serum conditions, they cannot be called senescent. The definition of a senescent cell is an irreversible exit from the cell cycle. The authors use b-galactosidase activity as a marker of senescence, but it is well established that lysosomal dysfunction can lead to increased lysosome biogenesis via TFEB pathway and thus increased bGal activity. In order to comprehensively establish whether these cells are senescent, I would like to see more characterisation of the cells, for example cell growth curves, expression of p16/p21, senescence-associated heterochromatin foci, nuclear size, cell size, SASP (see below) or similar.

The authors show that mTORC1 activity is low in *ctsd*^{-/-} cells despite the fact that they contain higher (to my eye) levels of many amino acids. Coupled with the data in Figure 5c, where the authors show that even acute amino acid starvation cannot restore mTORC1 to the lysosomal surface, it seems there is a fundamental disconnect between amino acid availability and mTORC1 activity. This is an extremely interesting observation, which I appreciate is outside the scope of this study to take any further, however I remain unconvinced that mTORC1 plays a large part in your model. mTORC1 activity is reduced in *ctsd*^{-/-} cells regardless of serum culture conditions but autophagy and lysosomal expansion is significantly upregulated in low serum conditions, does this mean that autophagy/lysosome biogenesis is upregulated via an mTORC1-independent mechanism in starvation? The flux experiments (see below) will help answer this question. If you restore mTORC1 activity (i.e. OE Rheb/active RabA/B), do you rescue the proliferation defect? I remain unsure whether other starvation responses, in addition to mTORC1 may be driving your phenotypes. The authors also need to discuss how their observations of reduced mTORC1 activity can be aligned with the current understanding that mTORC1 activity drives senescence-associated phenotypes such as increased cell size and SASP.

It's not possible to assess autophagy without carrying out flux experiments, especially in a complicated model such as *ctsd*^{-/-} where degradation is likely to be perturbed. Indeed the authors show that aspartic protease activity is perturbed whereas cysteine protease activity is not, thus presumably there is at least partial perturbation of lysosomal degradation. The experiments in Figure 4 and the associated extended figure need to be done in the presence of bafilomycin or similar.

Specific points:

- Is there an increase in senescence in vivo during the latency period, as the authors postulate in the discussion?
- The title for figure 4 is 'CTSD deficiency increases the autophagic flux in short-term starved tumor cells'. This is misleading since there are no measurements of flux. These experiment must be done in order to conclude how autophagy is altered in *ctsd*^{-/-} cells. The increase in LC3-II seen in 4a could be either a result of increased autophagosome production (as suggested by increased mRNA) or perturbed degradation. From looking at the blots, it doesn't look like autophagy is induced as much upon starvation in *ctsd*^{-/-} cells compared to controls.

- The blots in figure 5 a (and thus S3) are not convincing, could they be replaced with another representative blot? Can the authors clarify which band p70S6K? The loading for lane 1, to which everything is normalised is much lower than the others. This is an important point for the paper (see above) and the mechanism that the authors propose.
- Figure 7 is very interesting; can you show protein levels of phospho/total CREB? Have the authors considered a recent paper from Jewell et al, eLife indicating cAMP inhibits mTOR. Could mTORC1 being actively downregulated in these cells?
- I would not expect the authors to do extensive more experiments re. CREB but I think the overall conclusion of the paper would be made significantly stronger if the authors were to demonstrate CREB is important for 'senescence' escape. Could the authors KO/inhibit CREB and study re-entry into cell cycle described in Figure 6.

Specific technical points:

- Blot in 2b; why is there just one band for CTSD
- Please include error bars in Figure 2 (Day 1 recombination is 80% in a but only 60% in c)
- 2e is the control just one data point or are all data points normalised?
- I may misunderstand but to my eyes, 3e is not consistent with the authors conclusion. It is not clear to me what the individual data points are and there is no increase in acidic structures labelled high upon starvation. Thus the large vacuoles are unlikely to be classical, functioning lysosomes? Is there any statistical different between the profile in 10% versus 1% fcs?
- 3f: can you quantify this blot. It is hard to interpret as starvation leads to a decrease in Cath L and tubulin but not Lamp(?). Can the authors comment on why this may be if there is upregulation of CTSL mRNA and the authors conclusion is that there are more lysosomes?
- In Figure 3, the authors measure aspartic protease activity and cysteine protease activity. Could the authors total lysosome activity?
- Staining for LC3 in 4e is not convincing, it needs to be repeated. There are several standard antibodies in the field that work very well for IF. Alternatively, when the authors include flux experiments (see above), this panel may be redundant.
- 6e; can you add the specifics of this assay in the methods? How were 'living cells' identified by FACs?

Reviewer #2 (Remarks to the Author):

In this article titled 'Cathepsin D deficiency in mammary epithelium transiently stalls breast cancer by interference with mTORC1 signaling' the authors Ketterer et al. have used a conditional knock-out mouse model for Cathepsin D in the mammary epithelium in a MMTV-PyMT breast cancer model. The authors demonstrate that in the knock-out mouse model tumor growth is initially severely retarded and they demonstrate that this is due to suppressed mTORC1 signaling. They show that once the Cathepsin D deficient tumors do start growing they do so without significant input from the mTOR pathway, suggesting that Cathepsin D is important for the proper functioning of the mTORC1 pathway.

The article is very well written, and the authors use a very elegant mouse model to demonstrate that knock-down of Cathepsin D in the tumor cells rather than in the myeloid compartment is important for the tumor growth. This answers the question that has been posed before whether the secreted Cathepsin D is important for tumor formation or its intracellular functions in the tumor cells. They also show how the Cathepsin D knock-out cells can potentially escape the proliferation arrest by upregulating the genes required for EMT, through CREB activation.

The article is interesting, the experiments are generally well executed and controlled, and the authors do create an interesting narrative as to how the tumor cells first cease from growing and later on circumvent the growth arrest. The only drawback of this study is that the mechanistic links between all the observations are somewhat weak, and the conclusions drawn seem quite premature, especially if some of the established links are not experimentally tested. The authors acknowledge this and state that to understand all these connections and mechanistic links, one needs to dive deeper into these data. However, I feel that at least some of these connections could be experimentally tested, to solidify the

authors conclusions. Apart from that I have only few major concerns and suggestions for how to improve this manuscript.

Major concerns:

Figure 2: The actin in blot in 2B is missing, also could you add the molecular size bars to all the Western blots.

Figure 3: The more epithelial nature of the knock-out cells is quite obvious already in the phase image in Figure 3, given that later on the authors claim that there is an EMT state change during their re-growth it would be appropriate to show some EMT markers for these cells, both +/+ and -/- cells in 10%, 1% early stage and late stage cultures (e.g. N-Cadherin, E-Cadherin, Vimentin). In this line, if you treat cells with TOR inhibitors, do they become more epithelial?

Is the lysosome biogenesis increased in the -/- knock-out cells in 1% serum, compared to the wild-type cells? This could explain the increased lysosomal content. The authors speculate this, but there is not really much data to support this.

Figure 4: The data in figure 4 to me suggests that there is a larger autophagic flux in the 10% serum, and that in 1% they both do similarly, one would assume that the differences in the amino acid flux in this case would be more different in the 10% serum condition, yet the targeted metabolomics analysis was done in 1% serum. One could use dialyzed serum to take reduce the amino acids coming from serum and analyze whether the differences in these conditions are more significant. This is important given the dual role of autophagy in early vs late stage tumorigenic. One would speculate that in the early stages when the tumor still have access to vasculature, the autophagy wouldn't be active (and if I remember correctly the early stage autophagy is tumor suppressive) vs later stage the autophagy is more necessary, and shown to increase tumor growth. This would suggest that the Cathepsin D knock-out is tumor suppressive in early stages due its induction of autophagy even in the presence of full serum and other growth factors.

Figure 5a: The blot for p-S6K1 is missing, perhaps a conversion issue when generating the PDF?

Figure 5C: It is somewhat unexpected that the LAMP/TOR co-localization doesn't really change at all in the wildtype cells between the 10%, 1% and the amino acid stimulation, one would assume that at least with the acute amino acid stimulation vs long term 1% serum starvation, you should see bigger mTOR co-localization with LAMP1, however in all the graphs the col-localization levels remain the same. Therefore, perhaps it would be best to include the western blot data for p-S6K1 to support these data regarding mTORC1 activation.

Figure 7: The authors should also validate the p-CREB data with traditional western blots/IF and nuclear staining. Since the data suggest, and the authors speculate that this proliferation escape is due to increased TGFbeta signaling, and EMT, they should show that the EMT markers change in the way they suggest. Also if TGFbeta is involved, the authors should stimulate the early stage Cathepsin D -/- cells with TGFbeta to investigate whether this allows them to start proliferating faster, and escape the senescence block.

Reviewer #3 (Remarks to the Author):

Interesting and well written study that comprehensively investigates the role of Cathepsin D in breast cancer and find a key link to deficiency in the mTORC1 signaling pathway.

The authors further make the surprising finding that cathepsin-D deficient breast cancers can escape the blockade after a significant delay. They demonstrate that mTorc1 functionality remains impaired, but that the tumour can be reprogrammed to proliferate - probably via CREB.

A major question that is still unclear from my perspective is in regards to how and why Cathepsin D is so critical for assembly of the mTorc1 complex on the lysosomal surface. Is this linked to protease activity? Do cathepsin D inhibitors also affect mTorc1 assembly? More discussion about the possible reasons behind this unexpected effect would be important.

Point-by-point response to reviewers' comments on manuscript NCOMMS-19-26147-A "Cathepsin D deficiency in mammary epithelium transiently stalls breast cancer by interference with mTORC1 signaling"

Reviewer #1:

We would like to thank Reviewer #1 for critically reading and commenting on our manuscript. In the following, we address the comments point by point. Please note that our answers refer to the new figure numbering.

- 1) If the *ctsd*^{-/-} cells proliferate, even slowly, in low serum conditions, they cannot be called senescent. The definition of a senescent cell is an irreversible exit from the cell cycle. The authors use b-galactosidase activity as a marker of senescence, but it is well established that lysosomal dysfunction can lead to increased lysosome biogenesis via TFEB pathway and thus increased bGal activity. In order to comprehensively establish whether these cells are senescent, I would like to see more characterisation of the cells, for example cell growth curves, expression of p16/p21, senescence-associated heterochromatin foci, nuclear size, cell size, SASP (see below) or similar.

We would like to thank the reviewer for this comment, which induced a number of additional interesting experiments. The increase of acidic β -galactosidase activity, which is often considered as a senescence marker, together with an increased sensitivity to the so-called "senolytic" agent Navitoclax, prompted us to explain the proliferation defect seen in starved *Ctsd*^{-/-} cells by senescence. But of course, in the original submission we have also shown that this "senescent" cell state is reversible upon long-term culture. By examining the literature more carefully we found that our data would fit the term cellular quiescence that describes a reversible cell cycle arrest and is associated with the inhibition of the mTOR pathway (Terzi, M. Y., et al. The cell fate: senescence or quiescence. *Molecular Biology Reports* vol. 43 1213–1220 (2016)). As suggested, we characterized the proliferation defect of *Ctsd*^{-/-} cells further: Quantitative RT-PCR for the cyclin-dependent kinase inhibitors *Cdkn1a/p21* and *Cdkn2a/p16* (**new Fig. 3f**) showed a significant increase of *p16* mRNA indicating a cell cycle arrest. However, secretion of the major SASP cytokine IL-6 was not increased in cell-conditioned media from these cells (**new Fig. 3e**). Together with the reduced mTORC1 activity of *Ctsd*^{-/-} cells, these data argue rather for quiescence instead of senescence and we exchanged the terms accordingly in our manuscript.

- 2) The authors show that mTORC1 activity is low in *ctsd*^{-/-} cells despite the fact that they contain higher (to my eye) levels of many amino acids. Coupled with the data in Figure 5c, where the authors show that even acute amino acid starvation (did you mean stimulation?) cannot restore mTORC1 to the lysosomal surface, it seems there is a fundamental disconnect between amino acid availability and mTORC1 activity. This is an extremely interesting observation, which I appreciate is outside the scope of this study to take any further, however I remain unconvinced that mTORC1 plays a large part in your model. mTORC1 activity is reduced in *ctsd*^{-/-} cells regardless of serum culture conditions but autophagy and lysosomal expansion is significantly upregulated in low serum conditions, does this mean that autophagy/lysosome biogenesis is upregulated via an mTORC1-independent mechanism in starvation? The flux experiments (see below) will help answer this question. If you restore mTORC1 activity (i.e. OE Rheb/active RabA/B), do you rescue the proliferation defect? I remain unsure whether other starvation responses, in addition to mTORC1 may be driving your phenotypes.

Yes, there is a disconnection between amino acids availability and mTOR activity in *Ctsd*^{-/-} cells. We think that this disconnect is the key to the cell and animal phenotypes we observe. Regarding the connection of mTOR activity and autophagy/lysosomal biogenesis we would like to explain that the state of the art is, that active mTORC1 suppresses autophagy and lysosomal biogenesis. It does so (for instance) by phosphorylating TFEB, which keeps the transcription factor in the cytosol, and by adding inhibitory phosphates to the autophagy initiator kinase ULK1. Inhibition of mTORC1 (e.g. by starvation-induced dissociation from the lysosome) leads to exhaustion of the suppressor signals and to activation of autophagy and lysosomal biogenesis. We checked phosphorylation of p38 MAPK as an alternative inducer of lysosome biogenesis, but found no significant effect (**new Fig. 6a**). For CTSD knockout we show reduced phosphorylation of important targets downstream of mTORC1 (P70S6K and – **newly – ribosomal subunit S6**; **Fig. 6b, c**) and more acidic vesicles. This is all in line with the reduced mTORC1 activity in the *Ctsd*^{-/-} cells.

What remains is the disconnect described above – even when we re-stimulate amino acid-starved CTSD knockout cells with amino acids there is no re-association of mTOR with the lysosome (**Fig.6d-g**, for which we have repeated the experiments to provide a solid basis for the quantification). A great deal of effort in this revision was devoted to the examination of players that tether mTOR to the lysosome, and now we describe this set of experiments in the result section (**new Extended Data Fig. 5**). The results are in the section “Deregulated mTORC1 signaling in CTSD-deficient breast cancer cells” in the following passage:

*Next, we aimed to address effects of CTSD deficiency on factors known to tether mTOR to the lysosome and to investigate mTORC1 assembly. Lysosomal localization of mTOR has been shown to depend on RagGTPases that are recruited to the lysosome by a pentameric complex called Ragulator³³. In response to sufficient amino acid levels the RagGTPases bind the mTORC1 component Raptor, thereby placing mTOR at the lysosomal surface. First, we investigated protein levels of the two interaction partners Raptor and RagC. Comparing *Ctsd*^{-/-} to *Ctsd*^{+/+} cells, there was no lack of any of the two proteins (Extended Data Fig. 5a). Second, we assessed the assembly of mTORC1. To this end, the mTORC1 component mLST8 was overexpressed as an EGFP-fusion protein in *Ctsd*^{+/+} and *Ctsd*^{-/-} cells. Using GFP-Trap beads, we pulled down the EGFP-fusion protein and its interaction partners. EGFP-mLST8 pull-downs showed immunoreactivity with mTOR and Raptor antibodies, while the EGFP-only control showed no signal (Extended Data Fig. 5b). The EGFP-mLST8 pull-down was successful in both *Ctsd* genotypes, indicating a correct mTORC1 assembly in *Ctsd*^{-/-} cells. Next, mTORC1-interacting proteins at the lysosomal membrane were investigated for their expression levels and localization in *Ctsd*^{+/+} and *Ctsd*^{-/-} cells. Protein levels of Rheb, the small GTPase important for growth factor signal transmission and mTOR activation, as well as the Ragulator component Lamtor4 were equally present (Extended Data Fig. 5c). Furthermore, LAMTOR4 co-localization with LAMP1 did not differ between *Ctsd*^{+/+} and *Ctsd*^{-/-} cells (Extended Data Fig. 5d).*

*Together, these data indicate that lysosomes of CTSD-deficient cells are equipped with important mTORC1 interaction partners and principal mTORC1 assembly is not impaired by CTSD deficiency. However, mTORC1 association at the lysosomal surface is substantially impaired in absence of CTSD. This suggests that the accumulation of morphologically aberrant lysosomes is the basis for the sustained displacement of mTOR from the lysosome in *Ctsd*^{-/-} cells.*

In addition, we expressed RagA^{Q66L}, a GTP hydrolysis mutant of RagA, in *Ctsd*^{+/+} and *Ctsd*^{-/-} cells in an attempt to rescue mTORC1 activity. This did not consistently induce proliferation of starved *Ctsd*^{-/-} cells. We felt that it would need extensive effort to perform conclusive experiments using this approach, and as we are unsure whether this would contribute to a significantly enhanced

understanding of the molecular mechanisms underlying the *Ctsd*^{-/-} cell phenotype we would not like to present these data in the present manuscript.

- 3) The authors also need to discuss how their observations of reduced mTORC1 activity can be aligned with the current understanding that mTORC1 activity drives senescence-associated phenotypes such as increased cell size and SASP.

We agree that mTOR activity is important for the senescence-associated secretory phenotype and increase in cell size. Quantifying the amount of the SASP factor IL-6 in cell-conditioned media revealed a rather reduced IL-6 secretion from *Ctsd*^{-/-} cells (**new Fig. 3e**). This does not only remove possible inconsistencies between mTORC1 inactivation and SASP factor production, but also let us change 'senescence' to 'quiescence' in the manuscript (see also above).

- 4) It's not possible to assess autophagy without carrying out flux experiments, especially in a complicated model such as *ctsd*^{-/-} where degradation is likely to be perturbed. Indeed the authors show that aspartic protease activity is perturbed whereas cysteine protease activity is not, thus presumably there is at least partial perturbation of lysosomal degradation. The experiments in Figure 4 and the associated extended figure need to be done in the presence of bafilomycin or similar.

We would like to thank the reviewer for this comment. As suggested, we repeated the analysis of autophagy in presence of Bafilomycin A1 or Pepstatin A and E64d (**new Extended Data Fig. 2**). The results are in the section "CTSD deficiency results in macroautophagy" in the following passage (**section on Bafilomycin and protease inhibitors in bold**):

Ctsd^{-/-} PyMT cells showed an increased LC3-II/LC3-I ratio as well as an induction of Lc3a mRNA expression when cultured in presence of 10% FCS (Fig. 5a-c). These differences did not reach significance in FCS-starved medium, most likely because of the induction of autophagy in *Ctsd*^{+/+} control cells. Together these data suggest that autophagy is induced and also successfully executed in *Ctsd*^{-/-} cells. As a further indicator for that, protein levels of the autophagy cargo receptor p62 were reduced upon FCS starvation in *Ctsd*^{+/+} and *Ctsd*^{-/-} cells, probably due to p62 degradation in the autolysosome (Fig. 5a). Of note, the *Ctsd* genotype did not affect p62 protein levels, despite an upregulation of p62 mRNA in *Ctsd*^{-/-} cells (Fig. 5d). **In order to further assess autophagic flux, cells in FCS- and amino acid-free medium were treated with Bafilomycin A1 or a combination of Pepstatin A and E64d to block lysosomal acidification and protease activity, respectively (Extended Data Fig. 2). Both treatments led to an increase of the LC3-II/LC3-I ratio followed by the accumulation of p62 protein, independent of the *Ctsd* genotype. This clearly indicates that the proteolytic degradation of autolysosomal content in *Ctsd*^{-/-} cells is intact, which is also supported by the maintained global proteolytic capacity of CTSD-deficient PyMT cells reported in figure 4f. Consequently, the increased LC3-II/LC3-I ratio seen in *Ctsd*^{-/-} cells under FCS starvation is not a result of perturbed degradation but represents an increase of autophagic flux.** Furthermore, immunofluorescent co-staining of autophagosomal LC3 and lysosomal LAMP1 showed equal co-localization in *Ctsd*^{+/+} and *Ctsd*^{-/-} cells under normal and starving conditions (Fig. 5e). In synopsis, the data suggest an unperturbed formation of autolysosomes and an increased autophagic flux with the generation of free amino acids in *Ctsd*^{-/-} cancer cells.

Specific points:

- Is there an increase in senescence in vivo during the latency period, as the authors postulate in the discussion?

Key findings of this manuscript are the reduced development of mammary adenomas in 8-week-old CTSD-deficient MMTV-PyMT mice (Fig. 1a, b) and a delay of more than 8 weeks in the occurrence of

first palpable tumors (Fig. 1c), an effect that is remarkably strong even when compared to the knockout of major oncogenes in the model. Acute deletion of CTSD from mammary cancer cells by Dox-induced expression of cre recombinase resulted in equally strong growth disadvantage in cell culture (Fig. 2 c, d) and upon injection into mouse mammary glands (Fig. 2e). We also provide evidence, that CTSD-deficient untransformed mammary epithelium is capable of executing its main function, i.e. producing milk to raise the offspring successfully (Ext. Data Fig. 1d). CTSD deficiency is also not affecting estrogen receptor levels in the mammary epithelium (Ext. Data Fig. 1f). We assume that the PyMT oncogene-driven activation of the PI3K/AKT/mTOR signaling axis is prerequisite for the observed phenotype. This activation of PyMT occurs in B6 mice (our background) at around 6 weeks of age, i.e. later than in the often used FVB/n background. Thus, there is only a small time window to study the effect of CTSD deficiency in very early stages of PyMT-induced transformation. We assume that our detailed investigations on mTORC1 localization and activity in cells translate to the *in vivo* situation. As discussed before, we agree that senescence might not be the right term to describe the cell state in the tumor latency period. Staining the whole mounts presented in Fig. 1a for endogenous acidic β -galactosidase activity yielded high backgrounds leading to ambiguous results. We agree that detailed investigations of the latent stages would be very interesting. But we feel this will be a future project in its own right, also requiring additional animal studies, yet not covered by our ethics approval.

- The title for figure 4 is ‘CTSD deficiency increases the autophagic flux in short-term starved tumor cells’. This is misleading since you there are no measurements of flux. These experiment must be done in order to conclude how autophagy is altered in *ctsd*^{-/-} cells. The increase in LC3-II seen in 4a could be either a result of increased autophagosome production (as suggested by increased mRNA) or perturbed degradation. From looking at the blots, it doesn’t look like autophagy is induced as much upon starvation in *ctsd*^{-/-} cells compared to controls.

We thank the reviewer for this comment, we carefully revised the section on autophagy. As suggested, we performed and include autophagic flux experiments, also using Bafilomycin A1 and lysosomal protease inhibitors. They are described above (see answer to point 4) and confirm our statement of increased autophagic flux in starved *Ctsd*^{-/-} cells.

- The blots in figure 5 a (and thus S3) are not convincing, could they be replaced with another representative blot? Can the authors clarify which band p70S6K? The loading for lane 1, to which everything is normalised is much lower than the others. This is an important point for the paper (see above) and the mechanism that the authors propose.

It proved technically difficult to get signals for “total” and “phospho” S6 Kinase on the same membrane by antibody “stripping”. Therefore, we blotted two aliquots of the protein sample on two membranes, first normalized the band to the loading control (here TUBA) and subsequently calculated to phospho-/total ratio (this procedure is now more clearly described in methods). The experiment was repeated three times with independently thawed batches of cells and quantification is provided in Fig. 6c. The p-P70S6K blot now shows the relevant band. Because the status of S6K is indeed very important for our arguments, we performed and added an analysis of the main target of P70S6K, namely the phosphorylation of the ribosomal subunit S6 (**new Fig. 6b**). In line with our previous findings, S6 phosphorylation is reduced in *Ctsd*^{-/-}.

- Figure 7 is very interesting; can you show protein levels of phospho/total CREB? Have the authors considered a recent paper from Jewell et al, eLife indicating cAMP inhibits mTOR. Could mTORC1 being actively downregulated in these cells?

Response to Creb-Question: We quantified protein levels of phosphorylated and total CREB in whole cell lysates and in the nucleus (**new Extended Data Fig. 10a-c**). The results are described in the

section “CTSD-deficient cells rewire oncogenic cellular signaling and gene expression upon long-term FCS starvation” as follows:

*We validated the differential phosphorylation of CREB by different means. Quantification of total and phosphorylated CREB by alphaLISA revealed an increase of the p-CREB/CREB ratio in *Ctsd*^{-/-} cells during long-term starvation, while in *Ctsd*^{+/+} cells it was rather mildly decreased (Extended Data Fig. 10a). As the amount of phosphorylated CREB in the nucleus is decisive for its function as a transcriptional activator, we went further and performed immunofluorescent stainings in 1% FCS and 1% FCS LT *Ctsd*^{+/+} and *Ctsd*^{-/-} cells (Extended Data Fig. 10b). Quantification confirmed the increase of the nuclear p-CREB/CREB ratio in *Ctsd*^{-/-} cells, while this ratio remained unaffected in long-term FCS-starved *Ctsd*^{+/+} cells (Extended Data Fig. 10c).*

Response to integration of the Jewell et al findings: Thank you for making us aware of this interesting work. We now discuss this as follows:

*In a recent paper Jewell et al. report mTORC1 inhibition downstream of GPCR signaling through PKA-mediated Raptor phosphorylation⁴⁴. Concomitantly, PKA also phosphorylated CREB. However, in this report Raptor phosphorylation did not block mTORC1 association with the lysosome upon amino acid stimulation as we observed in *Ctsd*^{-/-} PyMT cells. In addition, our data show mTORC1 inactivation preceding the increase of CREB phosphorylation. Therefore, mTORC1 in CTSD-deficient PyMT cells is unlikely to be actively downregulated by PKA-mediated Raptor phosphorylation, while we cannot exclude a contribution of PKA signaling to CREB activation in the long term.*

- I would not expect the authors to do extensive more experiments re. CREB but I think the overall conclusion of the paper would be made significantly stronger if the authors were to demonstrate CREB is important for ‘senescence’ escape. Could the authors KO/inhibit CREB and study re-entry into cell cycle described in Figure 6.

We performed a metabolic activity assay on *Ctsd*^{+/+} and *Ctsd*^{-/-} cells in presence of the CREB inhibitor KG-501 (**new Extended Data Fig. 10d**). The corresponding result section is below. We also would like to point out, that it is very likely that many pathways may contribute to the long-term “resistance” of the cancer cells coping with cathepsin D deficiency. Unlike many studies in the field (that “cure” cancer in relatively short-term experiments) we actively asked the question “what happens in the long term”? For this we followed the primary MMTV-PyMT cancers until they grew despite the CTSD knockout and cultured CTSD knockout cells for extensive periods. Indeed, we found that cancer cells install mechanisms (such as CREB) to overcome the protease deficiency. Interestingly, this is different to (postmitotic) neurons. CTSD deletion in those cells leads to non-adapted neurodegeneration and death of mice as well as of affected human patients (see Ketscher A, Ketterer S, ..., Reinheckel T. Neuroectoderm-specific deletion of cathepsin D in mice models human inherited neuronal ceroid lipofuscinosis type 10. *Biochimie*. 2016 Mar;122:219-26).

*This difference in p-CREB levels translated into differential sensitivity to the CREB inhibitor KG-501. The EC₂₀, EC₅₀ and EC₈₀ values for KG-501 were reduced by approximately 20% in 1% FCS LT *Ctsd*^{-/-} compared to 1% FCS LT *Ctsd*^{+/+} cells (Extended Data Fig. 10d).*

*Taken together, the data provide evidence for higher CREB phosphorylation and a CREB-associated oncogenic gene expression pattern in long-term-starved CTSD-deficient PyMT cells. In consequence, *Ctsd*^{-/-} cancer cells are able to re-install growth in 1% FCS LT conditions despite the continued impairment of mTORC1 signaling. It remains to be investigated whether additional molecular pathways help cancer cells to adapt to the deficiency of this major lysosomal aspartic protease.*

Specific technical points:

- Blot in 2b; why is there just one band for CTSD

Different lysis and blotting conditions may change CTSD band patterns on Western blots. We repeated this experiment and the blot now shows the different bands for CTSD (see Fig. 2b).

- Please include error bars in Figure 2 (Day 1 recombination is 80% in a but only 60% in c)

We repeated the experiments and include now error bars in Fig. 2a, c and d. Also, we comment on the difference of recombination efficiencies in Fig. 2a versus Fig. 2c and d in the result section “Tumor cell-autonomous effects of CTSD deletion” in the following passage:

In order to achieve a balanced number of recombined and non-recombined cells in these assays, we seeded cells and allowed them to grow for 3 days before inducing recombination by adding Dox for 24 h. The cell cycle is desynchronizing during the three-day culture period. As cre-mediated recombination occurs preferentially during S-phase²⁶, this regimen achieved a lower proportion of recombined cells (i.e. 60%; Fig. 2c, d) as compared to Dox application immediately after seeding (i.e. 84% in Fig. 2a).

- 2e is the control just one data point or are all data points normalised?

To clarify this point, we specified the experimental setup in the result section “Tumor cell-autonomous effects of CTSD deletion” in the following passage:

In this experiment we injected a cell population containing 88% recombined CTSD-deficient cells independently into 6 recipient mice and analyzed the resulting tumors by flow cytometry detecting the mTmG-fluorescence. On average, the proportion of CTSD-deficient cells decreased to 52% in the outgrown tumors, while CTSD-competent cells rose from 12% to 48% of the tumor masses.

- I may misunderstand but to my eyes, 3e is not consistent with the authors conclusion. It is not clear to me what the individual data points are and there is no increase in acidic structures labelled high upon starvation. Thus the large vacuoles are unlikely to be classical, functioning lysosomes? Is there any statistical difference between the profile in 10% versus 1% fcs?

The individual data points represent 6 independent experiments (biological replicates). Each experiment is represented by an individual symbol (e.g. experiment 1 = square; experiment 2 = triangle etc.). During revision we added three independent experiments to increase statistical power. Indeed, we could establish that the intensity of acidic vesicle staining depends on both, the FCS concentration in the medium as well as on the *Ctsd* genotype. The result section “CTSD deletion induces a quiescent cell state and expands the acidic cell compartment” now reads:

*... flow cytometry using LysoTrackerTM Green was performed. The majority of *Ctsd*^{+/+} cells stained with medium intensity, both under normal and starving conditions, while *Ctsd*^{-/-} cells showed high LysoTrackerTM intensity, especially during starvation (Fig. 4a). A three-way ANOVA considering the factors “intensity of LysoTrackerTM”, “*Ctsd* genotype” and “FCS” revealed significant dependence of the acidic cell compartment on *Ctsd* ($p < 0.001$) and the amount of FCS in the culture medium ($p = 0.001$). The distribution of LysoTrackerTM intensities in starved *Ctsd*^{-/-} cells was highly reminiscent of the one of Torin-treated *Ctsd*^{+/+} cells, the positive control for autophagy induction and lysosomal biogenesis³⁰.*

Regarding the functionality of the lysosomes we would like to refer to the answer to the next questions.

- 3f: can you quantify this blot. It is hard to interpret as starvation leads to a decrease in Cath L and tubulin but not Lamp(?). Can the authors comment on why this may be if there is upregulation of CTSL mRNA and the authors conclusion is that there are more lysosomes?

&

- In Figure 3, the authors measure aspartic protease activity and cysteine protease activity. Could the authors total lysosome activity?

Answer to both questions:

We quantified the blots for LAMP1 and CTSL. Yet, we did not mean to use the amount of cathepsin L protein as a surrogate for the number of lysosomes (Lamp1 is a much better marker for these organelles). Our aim was to characterize the state of lysosomal proteolysis in the CTSD-deficient cells. We now extend on this topic by measuring total endolysosomal proteolysis (**new Fig. 4f**) in intact cells and show cathepsin B Western blots (**new Fig. 4i**) in addition to cathepsin L. The results are in the section “CTSD deletion induces a quiescent cell state and expands the acidic cell compartment” in the following passage:

*To test lysosomal proteolysis in intact PyMT cells, we loaded the acidic cell compartment with a quenched fluorogenic substrate by endocytosis. Degradation of this substrate by lysosomal proteases results in dequenching and recovery of the fluorescent signal. Blocking the endocytic substrate uptake by Cytochalasin D or preventing lysosomal acidification by Bafilomycin A1 reduced lysosomal proteolysis (Fig. 4f). However, total lysosomal proteolysis was comparable between *Ctsd*^{+/+} and *Ctsd*^{-/-} cells, with an increase under FCS starvation. Interestingly, individual members of the same protease class responded differently to CTSD deficiency and/or FCS starvation. For example, CTSL protein was always increased in *Ctsd*^{-/-} cells, but decreased in 1% FCS to a similar extent than in *Ctsd*^{+/+} cells, despite an upregulation on mRNA level (Fig. 4g, h). In contrast to that, CTSB was rather unaffected by CTSD deficiency, but became induced under starvation, independent of the *Ctsd* genotype (Fig. 4i). In summary, CTSD deficiency renders tumor cells quiescent, increases their lysosomal compartment under FCS starvation, and rewires the lysosomal proteolytic system.*

- Staining for LC3 in 4e is not convincing, it needs to be repeated. There are several standard antibodies in the field that work very well for IF. Alternatively, when the authors include flux experiments (see above), this panel may be redundant.

We included images of repeated experiments in Fig. 5e, and also have assessed autophagic flux by applying Bafilomycin and protease inhibitors as described in the response to general comment 4.

- 6e; can you add the specifics of this assay in the methods? How were ‘living cells’ identified by FACS?

We “borrowed” the FACS cell proliferation dye labelling assay from immunologists. Cells were stained with the non-toxic cell proliferation dye eFluor™ 670 which binds to primary amines of proteins. After staining, cells were washed to remove superfluous dye and replated. Cells were analyzed by flow cytometry after 72 h of culture. Dividing cells distribute the dye to their daughter cells (i.e. show decreasing fluorescence intensity), while non-proliferating cells retain the dye (i.e. show high fluorescence intensity). As controls for gate setting served unstained cells (negative population) and freshly stained cells (100% positive population). Before flow cytometry, dead cells were labeled by propidium iodine (PI) and PI-positive “events” were excluded from the analysis. The Materials and Methods section “Flow cytometry” reads as follows:

For label-retention assays eFluor™ 670 intensity in PI⁻ cells was divided into negative, low, and high to distinguish between fast-, slow-, and non-proliferating (label-retaining) cells, respectively.

In other experiments we used flow cytometry to detect “living cells”, i.e. in the Navitoclax treatment presented in Figure 3d. Here, living cells are defined as cells that do not bind Annexin V (early

apoptosis marker), nor the dye 7-AAD, which is (similar to PI) labeling DNA of permeabilized, late apoptotic or necrotic cells. We specify “living cells” now as Annexin-V⁻7-AAD⁻ cells in the figure legend of 3d. The Materials and Methods section “Flow cytometry” reads as follows:

For apoptosis assays the amount of Annexin-V⁻7-AAD⁻ (living) cells relative to DMSO-treated control and normalized to Ctsd^{+/+} cells is given.

Reviewer #2:

We would like to thank Reviewer #2 for critically reading and commenting on our manuscript. In the following, we address the comments point by point. Please note that our answers refer to the new figure numbering.

Major concerns:

1) Figure 2: The actin in blot in 2B is missing, also could you add the molecular size bars to all the Western blots.

We repeated the Western blot in Fig. 2b. Now, the loading control (TUBA) is equally distributed. Also, we added the molecular weight bars to all Western blots.

2) Figure 3: The more epithelial nature of the knock-out cells is quite obvious already in the phase image in Figure 3, given that later on the authors claim that there is an EMT state change during their re-growth it would be appropriate to show some EMT markers for these cells, both +/+ and -/- cells in 10%, 1% early stage and late stage cultures (e.g. N-Cadherin, E-Cadherin, Vimentin). In this line, if you treat cells with TOR inhibitors, do they become more epithelial?

Besides the cell morphology, the RNAseq data provided evidence for EMT in our original submission in long-term cultured CTSD knockout cells. We now validate and support this by quantitative RT-PCR and Western blots (**new Extended Data Fig. 9**). The result section “CTSD-deficient cells rewire oncogenic cellular signaling and gene expression upon long-term FCS starvation” states:

Indeed, 1% FCS LT Ctsd^{-/-} cells appeared less epithelial as compared to 1% FCS short-term Ctsd^{-/-} cells (compare Fig. 7e and Fig. 3a). The acquisition of a mesenchymal cell morphology in Ctsd^{-/-} cells following long-term starvation was accompanied by a significant increase of the mesenchymal markers N-Cadherin and Vimentin both on mRNA and protein level (Extended Data Fig. 9a-c)

Regarding mTOR inhibitors: Treating Ctsd^{+/+} cells with the mTOR inhibitor Torin is reducing phospho-S6K (Fig. 6b, c) as well as mesenchymal marker proteins such as Vimentin or N-cadherin (see figure below), almost reaching levels of Ctsd^{-/-} cells, for which we have shown that mTOR signaling is impaired. We did not attempt to treat CTSD knockout cells with Torin to further inhibit the mTOR pathway in these cells.

Figure for review only. Analysis of CDH2 and VIM protein levels by Western blot (with TUBA and ACTB as loading control) in 10% FCS, 1% FCS, 1% FCS LT and Torin-treated Ctsd^{+/+} and Ctsd^{-/-} PyMT cells.

3) Is the lysosome biogenesis increased in the -/- knock-out cells in 1% serum, compared to the wild-type cells? This could explain the increased lysosomal content. The authors speculate this, but there is not really much data to support this.

The term “lysosomal biogenesis” refers to (transcriptional) induction of lysosomal enzymes and other proteins that are decisive for lysosomal function. Autophagy and lysosomal biogenesis are

interconnected processes mainly because they are induced by similar cues (e.g. starvation-induced mTORC1 inactivation). In many parts of the manuscript we provide *in vivo* and cell culture evidences for both – increased expression of lysosomal proteins (Fig. 4b, c, g, h; Extended Data Fig. 3a) and induction of macroautophagy (Fig. 5a-e; **new Extended Data Fig. 2**; Extended Data Fig. 3b). In the revision, we focused on autophagy by showing an increased autophagic flux in *Ctsd*^{-/-} cells and on the state of the mTORC1 complex (**new Extended Data Fig. 2**; **new Extended Data Fig. 5**). We think we now provide various complementary evidence for increased autophagy and lysosomal biogenesis in CTSD knockout cells.

4)Figure 4: The data in figure 4 to me suggests that there is a larger autophagic flux in the 10% serum, and that in 1% they both do similarly, one would assume that the differences in the amino acid flux in this case would be more different in the 10% serum condition, yet the targeted metabolomics analysis was done in 1% serum. One could use dialyzed serum to take reduce the amino acids coming from serum and analyze whether the differences in these conditions are more significant. This is important given the dual role of autophagy in early vs late stage tumorigenic. One would speculate that in the early stages when the tumor still have access to vasculature, the autophagy wouldn't be active (and if I remember correctly the early stage autophagy is tumor suppressive) vs later stage the autophagy is more necessary, and shown to increase tumor growth. This would suggest that the Cathepsin D knock-out is tumor suppressive in early stages due its induction of autophagy even in the presence of full serum and other growth factors.

Thank you for this interesting discussion concerning autophagy. In fact, autophagy is known to suppress tumor initiation. Classic mouse genetic studies knocking out components of the autophagic machinery, including Atg7, Atg5, and Becn1, show that when autophagy is impaired, there is an increase in tumor initiation. Also gain-of-function mutations or amplifications in PI3K or AKT, or PTEN loss or silencing, which all activate mTOR and thereby inhibit autophagy, are common oncogenic alterations, suggesting a potential importance of suppressing autophagy during tumor initiation (review for full details: Santana-Codina N, Mancias JD, Kimmelman AC. The Role of Autophagy in Cancer. *Annu Rev Cancer Biol.* 2017;1:19-39).

Since mTOR is a crucial regulator of autophagy, we repeated and extended our analysis on mTOR/LAMP1 co-localization at the lysosomal membrane depending on the *Ctsd* genotype but also on nutritive conditions along the lines suggested in your comment (Fig. 6d, **new e, f, g**). Lysosomal localization of mTORC1 is prerequisite for the mTOR-mediated suppression of autophagy and lysosomal biogenesis. It does so (for instance) by phosphorylating TFEB, which keeps the transcription factor in the cytosol, and by adding inhibitory phosphates to the autophagy initiator kinase ULK1. Inhibition of mTORC1 (e.g. by starvation-induced dissociation from the lysosome) leads to exhaustion of the suppressor signals and to activation of autophagy and lysosomal biogenesis. The key observation is that mTOR is practically not localized (mTOR/LAMP1 co-localization correlation coefficients near zero) at the lysosome of *Ctsd*^{-/-} cells irrespective of the growth condition (Full medium with 10% FCS; Starvation medium without amino acids or FCS and low glucose; Re-addition of amino acids to the starvation medium) (Fig. 6d, **new e, f, g**). The result section “Deregulated mTORC1 signaling in CTSD-deficient breast cancer cells” states:

Co-localization of mTOR and LAMP1 in Ctsd^{+/+} cells was reduced in amino acid-starved conditions when compared to complete medium (10% FCS: r = 0.406; w/o FCS, AA: r = 0.241). Importantly, mTOR/LAMP1 co-localization was essentially lost in Ctsd^{-/-} cells (10% FCS: r = 0.081; w/o FCS, AA: r = 0.056). We were able to rescue mTOR/LAMP1 co-localization in amino acid-starved Ctsd^{+/+} cells by the addition of amino acids, but failed to do so in Ctsd^{-/-} cells (Fig. 6f, g).

This means that autophagy is always “on” in the CTSD knockout cells. Indeed, this might contribute to tumor suppression in early stages of PyMT-induced tumorigenesis (which we have observed in the CTSD knockout – Fig. 1a-c).

As tumors grow, they encounter nutrient shortage and other challenges (inflammation; oxidative stress). Under such conditions autophagy is typically switching to a tumor-promoting process (full details: Santana-Codina N, Mancias JD, Kimmelman AC. The Role of Autophagy in Cancer. *Annu Rev Cancer Biol.* 2017;1:19-39). This is related to what we point out in the second part of this manuscript. With time cells activated by a potent oncogene (such as the PyMT) find a way around the growth block imposed by impaired mTOR signaling. We demonstrate this by following the primary MMTV-PyMT cancers until they grew despite the CTSD knockout and cultured CTSD knockout cells for extensive periods during which they recovered their growth rates. It is tempting to speculate that autophagy supports the “recovery” of CTSD-deficient cancer cells. However, our data show normalization of the lysosomal compartment (Fig. 7e; Extended Data Fig. 6a) in long-term cultured *Ctsd*^{-/-} cells and our molecular screens imply rather mechanisms such as CREB activation instead of autophagy.

5)Figure 5a: The blot for p-S6K1 is missing, perhaps a conversion issue when generating the PDF?

The p-P70S6K Western blot is included in Fig. 6b.

6)Figure 5C: It is somewhat unexpected that the LAMP/TOR co-localization doesn't really change at all in the wildtype cells between the 10%, 1% and the amino acid stimulation, one would assume that at least with the acute amino acid stimulation vs long term 1% serum starvation, you should see bigger mTOR co-localization with LAMP1, however in all the graphs the col-localization levels remain the same. Therefore, perhaps it would be best to include the western blot data for p-S6K1 to support these data regarding mTORC1 activation.

As explained above, we repeated the experiments to achieve better statistical power and plotted the quantitative data in one graph using the same scale for all experiments, which allows better comparison. As expected and shown in Figure 6d-g, *Ctsd*^{+/+} (wild-type) cells reduce mTOR/LAMP1 co-localization upon starvation and recover it after re-stimulation with amino acids.

7)Figure 7: The authors should also validate the p-CREB data with traditional western blots/IF and nuclear staining. Since the data suggest, and the authors speculate that this proliferation escape is due to increased TGFbeta signaling, and EMT, they should show that the EMT markers change in the way they suggest.

We quantified protein levels of phosphorylated and total CREB in whole cell lysates and in the nucleus, and inhibited CREB by the compound KG-501 (**new Extended Data Fig. 10a-d**). The results are described in the section “CTSD-deficient cells rewire oncogenic cellular signaling and gene expression upon long-term FCS starvation” as follows:

*We validated the differential phosphorylation of CREB by different means. Quantification of total and phosphorylated CREB by alphaLISA revealed an increase of the p-CREB/CREB ratio in *Ctsd*^{-/-} cells during long-term starvation, while in *Ctsd*^{+/+} cells it was rather mildly decreased (Extended Data Fig. 10a). As the amount of phosphorylated CREB in the nucleus is decisive for its function as a transcriptional activator, we went further and performed immunofluorescent stainings in 1% FCS and 1% FCS LT *Ctsd*^{+/+} and *Ctsd*^{-/-} cells (Extended Data Fig. 10b). Quantification confirmed the increase of the nuclear p-CREB/CREB ratio in *Ctsd*^{-/-} cells, while this ratio remained unaffected in long-term FCS-starved *Ctsd*^{+/+} cells (Extended Data Fig. 10c). This difference in p-CREB levels translated into differential sensitivity to the CREB inhibitor KG-501. The EC₂₀, EC₅₀ and EC₈₀ values for KG-501 were*

reduced by approximately 20% in 1% FCS LT *Ctsd*^{-/-} compared to 1% FCS LT *Ctsd*^{+/+} cells (Extended Data Fig. 10d).

Taken together, the data provide evidence for higher CREB phosphorylation and a CREB-associated oncogenic gene expression pattern in long-term-starved CTSD-deficient PyMT cells. In consequence, *Ctsd*^{-/-} cancer cells are able to re-install growth in 1% FCS LT conditions despite the continued impairment of mTORC1 signaling. It remains to be investigated whether additional molecular pathways help cancer cells to adapt to the deficiency of this major lysosomal aspartic protease.

Regarding EMT we refer to our answer to question 2: Besides the cell morphology, the RNAseq data provided evidence for EMT in our original submission in long-term cultured CTSD knockout cells. We now validate and support this by quantitative RT-PCR and Western blots (**new Extended Data Fig. 9**). The result section “CTSD-deficient cells rewire oncogenic cellular signaling and gene expression upon long-term FCS starvation” states:

*Indeed, 1% FCS LT *Ctsd*^{-/-} cells appeared less epithelial as compared to 1% FCS short-term *Ctsd*^{-/-} cells (compare Fig. 7e and Fig. 3a). The acquisition of a mesenchymal cell morphology in *Ctsd*^{-/-} cells following long-term starvation was accompanied by a significant increase of the mesenchymal markers N-Cadherin and Vimentin both on mRNA and protein level (Extended Data Fig. 9a-c).*

8) Also if TGFβ is involved, the authors should stimulate the early stage Cathepsin D ^{-/-} cells with TGFβ to investigate whether this allows them to start proliferating faster, and escape the senescence block.

In our original submission we discussed (in the discussion section) TGFβ involvement based on suggestions of the literature. To follow this up by data we treated short-term FCS-starved CTSD knockout cells with TGFβ at concentrations we used for earlier work with PyMT cells (*Kern et al Mol Cancer. 2015 14(1):39.*) However, the CTSD-deficient cells did not survive this treatment (see figure below). Next, we tried a TGFβ inhibitor (Novartis compound SB431542). However, this compound did not result in a clear and consistent recovery of an epithelial morphology of the long-term-starved CTSD knockout cells. As those easy experiments did not support our idea of TGFβ involvement, we did not further follow the TGFβ path and deleted the corresponding speculative section from the discussion (which is now used to better explain the various experimental evidence added during revision).

Figure for review only. Representative pictures of 1% FCS (top) and 1% FCS LT (bottom) *Ctsd*^{-/-} PyMT cells treated for indicated time periods with rhTGFβ (2 ng/ml) or TGFβ type I receptor inhibitor (SB431542, 10 μM), respectively. Bars, 50 μm.

Reviewer #3:

We would like to thank the reviewer for the positive comments on our manuscripts. In the following, we address the comments point by point. Please note that our answers refer to the new figure numbering.

A major question that is still unclear from my perspective is in regards to how and why Cathepsin D is so critical for assembly of the mTORC1 complex on the lysosomal surface. Is this linked to protease activity? Do cathepsin D inhibitors also affect mTORC1 assembly? More discussion about the possible reasons behind this unexpected effect would be important.

Thank you for this comment. To experimentally address the issue we treated *Ctsd*^{+/+} PyMT cells with the protease inhibitor Pepstatin A which targets lysosomal aspartic proteases (Cathepsins D and E, Napsin A) of which CTSD is predominantly expressed in PyMT cells. We starved the cells by omitting FCS and amino acids and measured mTOR co-localization with the lysosomal membrane protein LAMP1. The **new extended data figure 4** now reveals that Pepstatin A-treated *Ctsd*^{+/+} (wild-type) PyMT cells have a significantly lower lysosomal mTOR localization as compared to starved cells without this protease inhibitor. We conclude, that the missing protease activity of CTSD is indeed involved in the observed molecular and phenotypic changes.

In addition, we focused many experiments of this revision on the investigations of CTSD deficiency toward mTORC1 components (along the line of comments from the other reviewers). In synopsis of our data and as suggested by you, we now extend the discussion on the impact of CTSD toward mTORC1 as follows:

We provide evidence that major components recruiting mTORC1 to the lysosome as well as the proteins that activate lysosome-associated mTOR are present at normal levels in Ctsd^{-/-} cells. We also show by mLST8-GFP pull-down experiments that mTORC1 is able to assemble properly in CTSD-deficient cells. To our surprise, mTOR co-localization with the lysosomal membrane protein LAMP1 was strongly reduced in CTSD-deficient cells, despite the presence of high levels of free amino acids. Also, experiments with amino acid starvation and re-addition of amino acids had very little effect on the disturbed association of mTOR with the lysosome. As mTORC1 tethering to the lysosome is a key determinant for its activity, we suspected impaired mTOR signaling in the CTSD knockout. Indeed, phosphorylation of a major mTORC1 target, namely the P70S6K, was reduced indicating impaired mTORC1 activity in CTSD-deficient tumor cells. As integrator of proliferation signals and the internal metabolic state of cells mTORC1 decides in favor of cell growth and division or metabolic adaptation, e.g. by induction of autophagy. Taken together, all our cellular and molecular readouts strongly suggest disturbed lysosomal mTORC1 localization and therefore a strongly reduced mTOR signaling. Up to now, we cannot explain how CTSD deficiency affects the assembly of the mTORC1 multiprotein complex at the lysosomal surface. As described above, there is no lack of amino acids or individual proteins required for mTORC1 assembly or activity. However, patients with inherited CTSD deficiency develop neuronal ceroid-lipofuscinosis type 10, which is characterized by an altered lipid metabolism in the postmitotic neurons^{7,8}. We assume that also in Ctsd^{-/-} cancer cells changes in lysosomal lipid composition hinder the fine-tuning of mTORC1 assembly at the lysosomal surface.

REVIEWERS' COMMENTS:

Reviewer #1 (Remarks to the Author):

I think the authors have made every attempt to address my points and while I think there are still some outstanding (and very interesting) questions, I think they are largely outside the original scope of the study which was the study the role of Cathepsin D in breast cancer progression. I have made some very minor comments below but am happy to recommend the manuscript for publication.

- I appreciate the canonical role of mTORC1 in inhibiting autophagy as the authors have pointed out, but my point was that mTORC1 is already inhibited in *Ctsd*^{-/-} cells in the presence of 10% serum (thus autophagy is increased), but serum starvation still significantly enhances autophagy at the mRNA and protein level. This suggests to me that it could be independent of mTORC1 but as mentioned above, I don't think this issue detracts from the paper as a whole.
- Can you quantify the new Fig.6a-I would interpret this as p-p38 being significantly reduced in *Ctsd* cells rather than no difference as stated in the text.
- I think the authors have done sufficient investigation in the defects in mTOR localisation, as it is probably outside the scope of the paper but did overexpression of active Rag rescue mTOR recruitment to the lysosome? If yes, this may argue against a lysosome-centric reason for displacement of mTOR and perhaps rather suggest changes in aa signalling i.e. via a cytoplasmic sensor. I would not expect the authors to do more experimental work but if they already have this piece of data, it would help clarify their conclusion to the data in Supplementary 5.
- Final sentence of new paragraph re. mTOR localisation to the lysosome-can you change this to 'that aberrant lysosomes may be the basis for the sustained displacement of mTOR from lysosomes in *Ctsd*^{-/-} cells'

Reviewer #2 (Remarks to the Author):

The reviewers have responded to this reviewers previous critiques satisfactorily. One additional point that I would like to make that could support the findings further, was a study published a few years ago (PMID: 27913436) where the authors showed that high CREB expression could re-initiate cell cycle in cells that were chronically treated with mTOR inhibitors. In these cells, mTOR activity remained very low, yet the cells were proliferating due to CREB presence, and fits with the authors findings.

Reviewer #3 (Remarks to the Author):

I am happy that the authors have responded to my concerns.

Reviewer #4 (Remarks to the Author):

Overall, the method and analyses relevant to WES and RNA-seq are appropriate. The only statement that may need to be re-considered was "In general, all tumors exhibited low numbers of mutations, which confirms previous reports showing highest genomic stability of PyMT tumors among all investigated breast cancer models." (Line 15-17 of Page 14) This is not surprising as many previous studies have demonstrated that overall tumors from GEMM models just don't have the complex genomic landscape as in human malignancies. For tumors that are known have high TMB, like lung cancers, tumors from GEMM have way lower TMB. Therefore, the data from the mice cannot be used to support low TMB in human tumors.

Point-by-point response to reviewers' comments on manuscript NCOMMS-19-26147-A "Cathepsin D deficiency in mammary epithelium transiently stalls breast cancer by interference with mTORC1 signaling"

REVIEWERS' COMMENTS

Reviewer #1 (Remarks to the Author):

I think the authors have made every attempt to address my points and while I think there are still some outstanding (and very interesting) questions, I think they are largely outside the original scope of the study which was the study the role of Cathepsin D in breast cancer progression. I have made some very minor comments below but am happy to recommend the manuscript for publication.

- I appreciate the canonical role of mTORC1 in inhibiting autophagy as the authors have pointed out, but my point was that mTORC1 is already inhibited in *Ctsd*^{-/-} cells in the presence of 10% serum (thus autophagy is increased), but serum starvation still significantly enhances autophagy at the mRNA and protein level. This suggests to me that it could be independent of mTORC1 but as mentioned above, I don't think this issue detracts from the paper as a whole.

We thank the reviewer for this comment. We agree that autophagy is further increased upon FCS starvation of *Ctsd*^{-/-} cancer cells (Fig. 5b). Remarkably, this was accompanied by a further reduced phosphorylation of P70S6 kinase, indicating a further reduced mTORC1 activity (Fig. 6c). As reduction of mTORC1 activity stimulates autophagy this finding supports our idea of mTORC1 inactivation being important for the observed phenotypes in *Ctsd*^{-/-} cells.

- Can you quantify the new Fig.6a-I would interpret this as p-p38 being significantly reduced in *Ctsd* cells rather than no difference as stated in the text.

"No difference" was written in the point-by-point response and was not part of the manuscript text. We apologize for this somewhat inaccurate phrasing. But of course- the scientific question is valid, interesting and requires accurate phrasing.

In line with the reviewer's inquiry on mTORC1-independent autophagy induction, we examined the phosphorylation state of p38 MAPK (Fig 6a). Increased p38 MAPK phosphorylation is known to be an alternative activator of autophagy. This is the quantification of the analysis:

We agree with the reviewer's comment that p38 MAPK phosphorylation is decreased in *Ctsd*^{-/-} cells and phospho-p38 MAPK appears to be increased in 1% FCS conditions. Indeed, the latter could contribute to the increase of autophagy observed in FCS-starved *Ctsd*^{-/-} cancer cells. However, in *Ctsd*^{-/-} cells there is still more autophagy and less p-p38 MAPK than in *Ctsd*^{+/+} cells. In the manuscript we revised the section as follows (changes in red):

Phosphorylation of p38 MAPK has been shown to upregulate the transcription of lysosome and autophagy genes through inhibition of the transcriptional repressor ZKSCAN3³². FCS starvation activated the stress-responsive p38 MAPK irrespective of the *Ctsd* genotype (Fig. 6a). However, in both FCS conditions, phosphorylation of p38 MAPK was markedly reduced in *Ctsd*^{-/-} cells as compared to *Ctsd*^{+/+} cells. Because these results cannot completely explain the increased autophagy of FCS-starved *Ctsd*^{-/-} cells, we next examined the mTORC1 signaling pathway.

- I think the authors have done sufficient investigation in the defects in mTOR localisation, as it is probably outside the scope of the paper but did overexpression of active Rag rescue mTOR recruitment to the lysosome? If yes, this may argue against a lysosome-centric reason for displacement of mTOR and perhaps rather suggest changes in aa signalling i.e. via a cytoplasmic sensor. I would not expect the authors to do more experimental work but if they already have this piece of data, it would help clarify their conclusion to the data in Supplementary 5.

We were asked if we could rescue the proliferation defect by restoring mTORC1 activity. As explained in the first point-by-point response, overexpressing RagAQ66L in *Ctsd*^{-/-} cells did not rescue the proliferation defect. We agree with the reviewer, that one explanation could be that RagAQ66L overexpression was not sufficient to assemble the mTORC1 properly at the lysosomal membrane. Yet, we felt that investigating this and other possibilities pose a project itself and therefore did not address this further during revision of this manuscript.

- Final sentence of new paragraph re. mTOR localisation to the lysosome-can you change this to 'that aberrant lysosomes may be the basis for the sustained displacement of mTOR from lysosomes in *Ctsd*^{-/-} cells'

We thank the reviewer for this suggestion and changed the sentence accordingly.

Reviewer #2 (Remarks to the Author):

The reviewers have responded to this reviewers previous critiques satisfactorily. One additional point that I would like to make that could support the findings further, was a study published a few years ago (PMID: 27913436) where the authors showed that high CREB expression could re-initiate cell cycle in cells that were chronically treated with mTOR inhibitors. In these cells, mTOR activity remained very low, yet the cells were proliferating due to CREB presence, and fits with the authors findings.

We would like to thank the reviewer for his efforts, positive comments and for making us aware of this work that fits our ideas on how the effect of the CTSD knockout is overcome by the cancer cells. We now cite the paper and discuss this as follows (changes in red):

One downstream target of these kinases specifically activated by phosphorylation in CTSD-deficient previously quiescent tumor cells was CREB. Overexpression of this transcription factor has been linked to tumorigenesis and resistance to Raf/MEK/ERK and PI3K/Akt inhibitors⁴². Interestingly, tumor cells with chronic PI3K/mTOR inhibition are able to resume proliferation through CREB stabilization⁴³. Furthermore, progression of hypoxic tumors in PyMT mice is accompanied by CREB activation⁴⁴. Based on its activating phosphorylation and transcriptional induction of its target genes, we think that CREB is important for CTSD-deficient tumor cells to overcome the proliferation block during starvation. In a recent paper Jewell et al.⁴⁵ report mTORC1 inhibition downstream of GPCR signaling through PKA-mediated Raptor phosphorylation. Concomitantly, PKA also phosphorylated CREB. However, in this report Raptor phosphorylation did not block mTORC1 association with the lysosome

upon amino acid stimulation as we observed in *Ctsd*^{-/-} PyMT cells. In addition, our data show mTORC1 inactivation preceding the increase of CREB phosphorylation. Therefore, mTORC1 in CTSD-deficient PyMT cells is unlikely to be actively downregulated by PKA-mediated Raptor phosphorylation, while we cannot exclude a contribution of PKA signaling to CREB activation in the long term.

Reviewer #3 (Remarks to the Author):

I am happy that the authors have responded to my concerns.

We would like to thank this reviewer for his efforts and positive comments.

Reviewer #4 (Remarks to the Author):

Overall, the method and analyses relevant to WES and RNA-seq are appropriate. The only statement that may need to be re-considered was "In general, all tumors exhibited low numbers of mutations, which confirms previous reports showing highest genomic stability of PyMT tumors among all investigated breast cancer models." (Line 15-17 of Page 14) This is not surprising as many previous studies have demonstrated that overall tumors from GEMM models just don't have the complex genomic landscape as in human malignancies. For tumors that are known have high TMB, like lung cancers, tumors from GEMM have way lower TMB. Therefore, the data from the mice cannot be used to support low TMB in human tumors.

We would like to thank the reviewer for this comment. With this statement we did not mean to stress the low mutational burden in tumors from MMTV-PyMT mice as compared to human tumors. PyMT expression activates the oncogenic PI3K/Akt signaling pathway, one of the most frequently upregulated pathways in human breast cancer (*TCGA*. *Nature*. 2012; 490: 61-70). Therefore, the MMTV-PyMT model is excellent for studying a human-relevant oncogenic breast cancer pathway. Regarding the genomic stability of murine breast cancer models, the PyMT model has been shown to be genetically very stable as opposed for instance to the *p53*^{-/-} /*BRCA1*^{-/-} mice, which show a high degree of genomic instability and are therefore suited to study this aspect of tumorigenesis (*Ben-David et al.* *Nat Commun*. 2016; 7: 12160).

An often raised question concerning the escape of CTSD-deficient tumors from the growth blockade was whether CTSD-deficient mammary epithelial cells acquire relevant somatic oncogenic mutations during the latency period. Therefore, we performed whole exome sequencing of breast cancers of CTSD-deficient and wild-type mice and corrected to non-tumor tissue (tail) from the corresponding animals. Firstly, this experiment confirmed the known genetic stability of the MMTV-PyMT cancers. Secondly, we could exclude the occurrence of new oncogenic mutations in CTSD-deficient cancers as compared to cancers from *Ctsd*^{+/+} animals. These data led us to the search for compensatory oncogenic pathways upregulated by transcriptional and/or posttranscriptional mechanisms.